# ppGpp functions as an alarmone in metazoa

Doshun Ito[1], Hinata Kawamura[1], Akira Oikawa[2], Yuta Ihara[1], Toshio Shibata[3], Nobuhiro Nakamura [1], Tsunaki Asano[4], Shun-Ichiro Kawabata[3], Takashi Suzuki[1] & Shinji Masuda [1✉]

Guanosine 3′,5′-bis(pyrophosphate) (ppGpp) functions as a second messenger in bacteria to adjust their physiology in response to environmental changes. In recent years, the ppGpp-specific hydrolase, metazoan SpoT homolog-1 (Mesh1), was shown to have important roles for growth under nutrient deficiency in *Drosophila melanogaster*. Curiously, however, ppGpp has never been detected in animal cells, and therefore the physiological relevance of this molecule, if any, in metazoans has not been established. Here, we report the detection of ppGpp in *Drosophila* and human cells and demonstrate that ppGpp accumulation induces metabolic changes, cell death, and eventually lethality in *Drosophila*. Our results provide the evidence of the existence and function of the ppGpp-dependent stringent response in animals.

[1] Department of Life Science and Technology, Tokyo Institute of Technology, Yokohama, Japan. [2] Faculty of Agriculture, Yamagata University, Tsuruoka, Japan. [3] Department of Biology, Faculty of Science, Kyushu University, Fukuoka, Japan. [4] Department of Biological Sciences, Tokyo Metropolitan University, Hachioji, Japan. ✉email: shmasuda@bio.titech.ac.jp

     1

Organisms must adjust their physiology in response to environmental changes. The stringent response is one of the most important starvation/metabolic control responses in bacteria and is controlled by the hyper-phosphorylated nucleotide guanosine 3′,5′-bis(pyrophosphate) (ppGpp)[1,2]. ppGpp accumulates in bacterial cells upon exposure to various stresses and the accumulated ppGpp functions as an alarmone that can alter transcription[3–6], translation[7,8], and certain enzymatic activities[6,9,10] to overcome a stress[3]. In *Escherichia coli*, ppGpp level is regulated by two distinct enzymes, namely RelA and SpoT[11]. Both RelA and SpoT catalyze pyrophosphorylation of GDP (or GTP) by using ATP to produce ppGpp[12]. However, only SpoT (and not RelA) hydrolyzes ppGpp[13]. The RelA/SpoT homologs (RSHs) are universally conserved in bacteria[14] and have pivotal roles in various aspects of bacterial physiology, including the starvation response[1,2], growth rate control[15], antibiotic tolerance[16], and darkness response[17]. RSHs are also found in eukaryotes, including the green algae *Chlamydomonas reinhardtii*[18] and land plant *Arabidopsis thaliana*[19], as well as metazoa, including humans and *Drosophila melanogaster*[14,20].

The RSH superfamily is classified into three types: long RSHs, small alarmone synthases (SASs), and small alarmone hydrolases (SAHs)[14]. Long RSHs have ppGpp synthase and hydrolase domains near their N terminus and a regulatory domain near their C terminus, which associates with variable interactors such as the ribosome[21] and acyl carrier proteins[22]. SASs contain a single ppGpp synthase and SAHs contain a single hydrolase domain[23]. SASs are found not only in the bacterial phyla *Firmicutes* and *Actinobacteria* but also in archaea and some eukaryotes[14]. SASs in *Firmicutes* have been well characterized and include YwaC/RelP and YjbM/RelQ[14,23,24]. YwaC and YjbM have weak and strong ppGpp synthesis activity, respectively, for GTP homeostasis control[6,23]. SAHs are found in some bacterial species and in metazoa[14]. Metazoan SpoT homolog-1 (Mesh1) is one class of SAH found in metazoa (Fig. 1a). Mesh1 hydrolyzes ppGpp in vitro with comparable efficiency to bacterial long RSHs[25,26]. The *Drosophila Mesh1* loss-of-function mutant (*Mesh1 LOF*) exhibits retarded growth, especially during starvation[25], suggesting that the ppGpp-dependent stringent response has been conserved in metazoa. However, this hypothesis is still under investigation, because no known ppGpp synthase has been identified, and the existence of ppGpp in metazoa has never been confirmed.

Many studies have been conducted to attempt to detect ppGpp from metazoan cells[25,27]. The first attempt to detect ppGpp in eukaryotic cells was in 1970, in which nucleotides in HeLa cells were universally labeled with [$^{32}$P]orthophosphate and the cell lysates were subjected to thin-layer chromatography; however, no ppGpp signal was detected, even after starvation[28]. Several years later, a report was published describing the presence of ppGpp along with other highly phosphorylated nucleotides in Chinese hamster ovary cells and baby hamster kidney cells[29]; however, this result could not be reproduced[30]. More recently, Sun et al.[25] attempted to measure endogenous ppGpp in *Drosophila* by use of the high-performance liquid chromatography (LC) and matrix-assisted laser desorption/ionization-time of flight (TOF) mass spectrometry (MS); however, no ppGpp signals could be detected[25]. Therefore, it has been assumed that ppGpp is absent or, if present, accumulates very low levels in Metazoa[25,27,29,30].

Recently, human Mesh1 (hMESH1) was shown to function as a cytosolic NADPH phosphatase[31]. The enzymatic efficiency ($k_{cat}/K_m$) of NADPH phosphatase activity of hMESH1 was calculated to be 14.4 mM$^{-1}$ s$^{-1}$, which is comparable to that of ppGpp hydrolase activity (9.46 mM$^{-1}$ s$^{-1}$)[25,31]. Expression of hMESH1 in human cells was upregulated by cystine deprivation or addition of ferroptosis-inducing erastin and hMESH1 overexpression (OE) resulted in lowered NADPH levels with increased sensitivity to ferroptosis[31], suggesting that the NADPH phosphatase activity of hMESH1 is important in the regulation of ferroptosis. On the other hand, the physiological significance of the ppGpp-specific hydrolase activity of Mesh1 is currently unclear.

Here we were able to detect ppGpp in *Drosophila* and human cells, and demonstrate that Mesh1 modulates the endogenous ppGpp concentration in vivo, indicating that ppGpp metabolism is conserved in metazoa.

## Results

**Detection of ppGpp in *Drosophila*.** To check for the existence of ppGpp in animals, we employed ultra-performance LC coupled with a tandem quadrupole MS (MS/MS)-based ppGpp quantification method, which we previously established for the characterization of ppGpp function in plants[32]. We extracted nucleotide pools from *Drosophila* late third instar larvae, in which *Mesh1* expression was higher than at other stages as demonstrated by northern blotting[25]. Using the guanosine tetraphosphate-specific MS/MS mode ($m/z$ transition from 602 to 159), three peaks were observed at 5.4, 6.2, and 6.4 min (Fig. 1b, blue), similar to what was observed for the detection of ppGpp in *Arabidopsis* tissues[32]. To determine which peak, if any, contained ppGpp in the *Drosophila* extract, we added ppGpp to the larvae extract (final concentration 16.7 nM) and subjected the sample to MS/MS analysis. The elution profile of the mixed sample revealed an increased area of the 6.4 min peak (Fig. 1b, orange), which matched the ppGpp standard peak (Fig. 1b, pink), strongly suggesting that it corresponds to ppGpp elution, as also seen in *Arabidopsis*[32]. As we fragmented the possible ppGpp elution for MS/MS detection, we could not collect separated nucleotides after LC to verify the identity of the ppGpp peak by other methods. Instead, we analyzed multiple fragmented ions derived from the elution and the ppGpp standard. The ppGpp standard and the 6.4 min elution peak of the *Drosophila* extract showed similar fragmented ion patterns at $m/z = 150$, 159, and 177, which correspond to guanine, deoxy-pyrophosphate, and pyrophosphate, respectively (Supplementary Fig. 1a, b), strongly suggesting that the 6.4 min peak represents the ppGpp elution. The 5.4 min peak was assigned as GTP (owing to cross-talk of the product ion) and the 6.2 min peak represented an unknown molecule (Supplementary Fig. 1a). It is noteworthy that a small amount of the unknown molecule could be detected in the ppGpp (Fig. 1b, pink) and GTP standards (Supplementary Fig. 1a), suggesting that it was also contaminated in the manufacturing process of the products. The unknown molecule is likely a guanosine tetraphosphate, as it could be passed through the first selected MS (MS1) at $m/z = 602$ and has a diphosphate moiety that could be detected in the second selected MS (MS2) at $m/z = 159$ after fragmentation in the tandem mass spectrometer. The pattern of fragmented ions of the unknown molecule is very similar to that of ppGpp (Supplementary Fig. 1b), suggesting that the structure of the unknown molecule resembles that of ppGpp. The intensity of the unknown peak varied with different developmental stages of *Drosophila* (Fig. 1b, c and Supplementary Fig. 2), implying the physiological function of the unknown molecule that may be involved in developmental processes. These results indicated that our LC-MS/MS-based ppGpp quantification method allows detection of ppGpp and an unknown molecule from *Drosophila* tissues with high sensitivity.

We also attempted to detect another bacterial alarmone, guanosine 5′-triphosphate 3′-diphosphate (pppGpp)[1,2] from *Drosophila*; however, no stable pppGpp elution peak could be observed using the LC-MS/MS-based ppGpp quantification

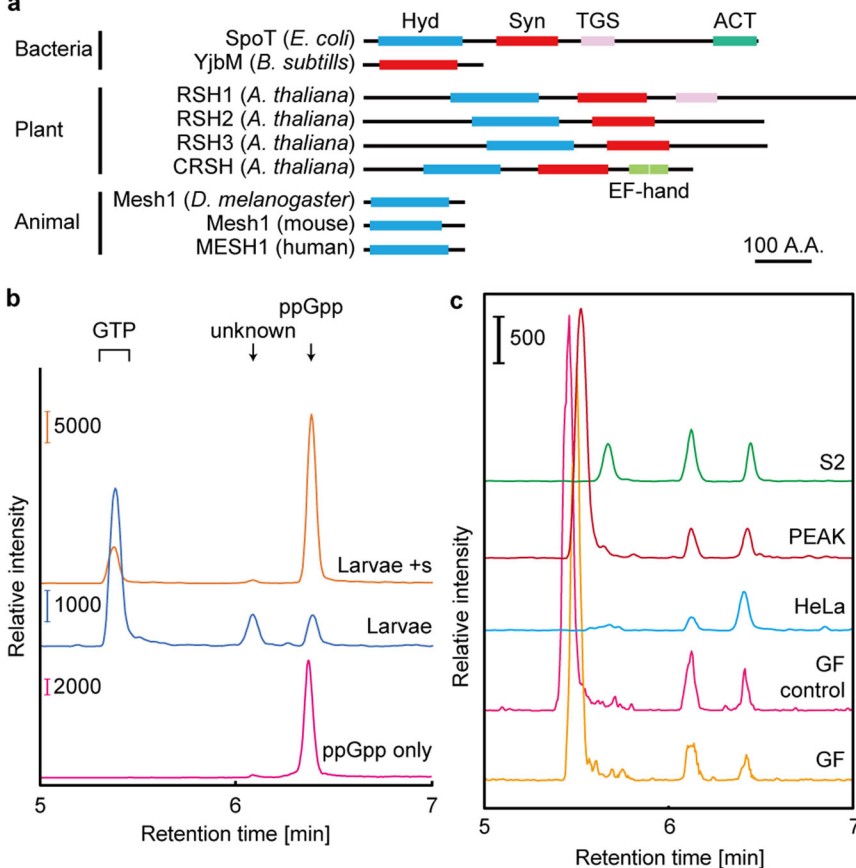

**Fig. 1 ppGpp detection in *Drosophila* and human cultured cell. a** Schematics of the primary structures of RSHs. A.A.: amino acids; ACT: aspartate kinase chorismite mutase TyrA domain; EF hand: $Ca^{2+}$ binding helix E and F motif; Hyd: (p)ppGpp hydrolase domain; Syn: (p)ppGpp synthase domain; TGS: threonyl-tRNA synthetase, GTPase, SpoT domain. **b** Multiple-reaction-monitoring chromatograms for guanosine tetraphosphate ions in extracts from $w^-$ third instar larvae. The symbol "+" denotes the exogenous addition of the ppGpp standard to the sample. **c** Detection of ppGpp from various cultured cells and germ-free (GF) flies. Guanosine tetraphosphate-specific MS chromatograms of nucleotide pools extracted from larvae grown under the normal (GF control) and GF conditions, human cultured HeLa cells, human cultured PEAKrapid cells, and *Drosophila* cultured S2 cells.

method described above, even when a pppGpp standard was used (for details, see "Methods"). This may result from reduced affinity of pppGpp for the reversed-phase column in the chosen solvent, compared to that of ppGpp, due to extra phosphate moiety of pppGpp. In future, selection of an appropriate ion-pair reagent and precise determination of the reagent concentration should be investigated for establishing the pppGpp-specific quantification method from *Drosophila* tissues.

We next tested stage-specific accumulation of ppGpp to investigate whether ppGpp levels modulate during *Drosophila* development. Specifically, using the MS/MS method, we quantified ppGpp concentrations from wild-type (WT) *Drosophila* ($w^-$) at various developmental stages: eggs (from the strain Canton-S: CS), late third instar larvae (male and female), pupae (day 2), virgins (male and female), and day 4 adults (male and female). ppGpp was detected in extracts from all stages (Fig. 1b and Supplementary Fig. 2a–d) at 50–250 pmol $g^{-1}$ fresh weight (FW) (Supplementary Table 1), which is comparable to that found in *Arabidopsis* (~100 pmol $g^{-1}$ FW)[32], and to the previous study that reported that ppGpp concentration in *Drosophila*, if any, should be less than 1000 pmol $g^{-1}$ FW[25]. Interestingly, the amount of ppGpp in pupae was greater than that at other stages (Fig. 2a, Supplementary Fig. 3a, and Supplementary Table 1). Given that *Mesh1* is highly expressed in larvae and suppressed in pupae[25], highly expressed Mesh1 in larvae seems to promote degradation of ppGpp and lowered *Mesh1* expression in pupae

could suppress ppGpp degradation. However, it is unknown whether ppGpp in *Drosophila* tends to be degraded or induced at each stage. The concentration of ppGpp (per FW) in female adults was slightly, but significantly, higher than in male adults (Fig. 2a); however, the relative proportion of ppGpp to GTP in females was ~50% less than that observed in males (Supplementary Fig. 3a). The GTP concentration in mating females was also higher (~3-fold) than in mating males (Supplementary Fig. 3b), although this concentration difference was not found in adult virgins (Fig. 2a and Supplementary Fig. 3a), suggesting that the increased ppGpp and GTP in mating females was derived from fertile eggs.

Bacterial cells contain a large amount of ppGpp (~470 nmol $g^{-1}$ FW)[33,34] and animals harbor symbiotic bacteria[35], suggesting that the ppGpp detected in *Drosophila* may have originated from such bacteria. To investigate this hypothesis, we quantified ppGpp in germ-free *Drosophila*, which lack bacterial symbionts[36]. Indeed, ppGpp was definitively detected in germ-free *Drosophila*, although the concentration was significantly lower than in non-germ-free control flies (Figs. 1c and 2b). These results indicated that although ~50% of the detected ppGpp might have originated from symbiotic bacteria (Fig. 2b), some ppGpp is synthesized in *Drosophila* by an unknown enzyme(s). To confirm this, we tested for ppGpp in extracts of germ-free human cells (HeLa and PEAKrapid) and *Drosophila* S2 cells in culture. The chromatograms showed peaks at the same retention times as those of late third instar larvae of

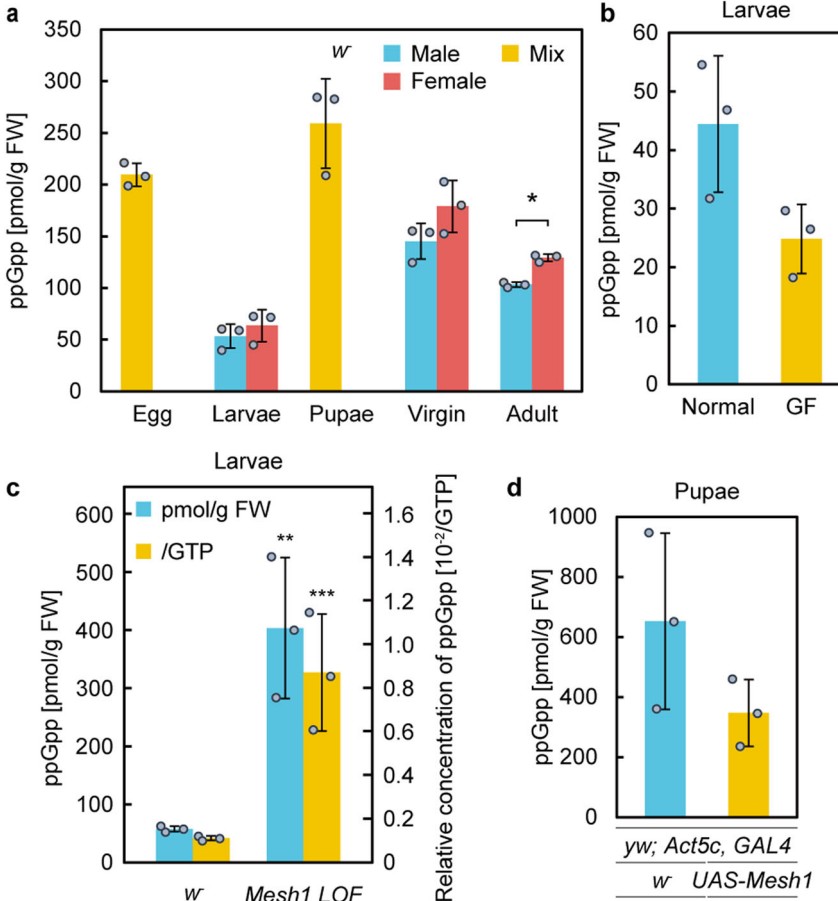

**Fig. 2 Concentration of ppGpp in various stages of development or transformants of *Drosophila*.** ppGpp levels per fresh weight (FW) in various stages in $w^-$ (**a**), $w^-$ instar larvae grown under normal and germ-free (GF) conditions (**b**), *Mesh1 LOF* (**c**), and *Mesh1 GOF* (**d**). Values represent the mean ± SD ($n = 3$ biologically independent samples). \*$p = 0.0008$, \*\*$p = 0.0384$ (compared to $w^-$), \*\*\*$p = 0.0392$ (compared to $w^-$); two-sided Student's *t*-test.

*Drosophila* (Fig. 1c). The ppGpp concentration in HeLa cells was ~40 pmol g$^{-1}$ FW and the relative concentration of ppGpp to GTP in both HeLa and S2 cells was ~$2.5 \times 10^{-3}$ (Supplementary Table 1), which is comparable to that determined in *Drosophila* larvae. These results indicated that ppGpp is present in metazoa and its concentration varies during development.

**Effect of Mesh1 function on ppGpp level.** Mesh1 expressed in *E. coli* exhibits ppGpp hydrolase activity[25,26]. However, Mesh1 activity in vivo has never been tested because ppGpp has not been detected in metazoa[25]. To verify that metazoan Mesh1 has ppGpp hydrolase activity, we quantified ppGpp in *Drosophila* extracts of *Mesh1 LOF* and the *Mesh1*-overexpressing transformant (*Mesh1 GOF: gain of function*)[25]. The chromatograms of ppGpp extracted from *Mesh1 LOF* had a sharp ppGpp peak (Supplementary Fig. 2e, f) and the concentration of ppGpp in larvae was significantly (~7-fold) higher than in WT (Fig. 2c and Supplementary Table 1). The *Mesh1 LOF* pupae contained ~3-fold more ppGpp than did WT (Supplementary Fig. 3c). To induce *Mesh1* expression in *Drosophila*, we applied a GAL4/UAS system[37]. We crossed *UAS-Mesh1* transformants with flies expressing GAL4 and used an actin promoter, *Act5c*, for systemic and constitutive OE of *Mesh1* (*Mesh1 GOF*). We then measured the amount of ppGpp in pupae. The concentration of ppGpp in *Mesh1 GOF* was approximately 347 pmol/g FW and in the control was 652 pmol g$^{-1}$ FW (Supplementary Table 1), suggesting that ppGpp degradation was enhanced in *Mesh1 GOF* compared to that in the control strain, although no clear significance was observed ($P = 0.21$, *t*-test)

(Fig. 2d). The ppGpp level in the control strain of *Mesh1 GOF* was higher than in $w^-$ (Supplementary Table 1). The reason for the difference is currently unclear; however, it might be caused by different genotypes. These results implied that Mesh1 has ppGpp hydrolase activity in vivo. We also found that GTP levels in *Mesh1 LOF* larvae and pupae were ~10% and 20% lower, respectively, than those in WT ($w^-$) (Supplementary Fig. 3d). Given that GTP synthesis is suppressed by ppGpp in Gram-positive bacteria[6], ppGpp may play a role in GTP homeostasis in metazoa, although GTP levels in *Mesh1 GOF* pupae were the same as those in WT (Supplementary Fig. 3e).

To further investigate Mesh1 and ppGpp function in *Drosophila*, we characterized the *Mesh1 LOF* mutant phenotype during hatching from larvae to pupae under starvation conditions. On rich medium (control), most larvae hatched at the second day after transfer and hatching rates from larvae to pupae were 82% for $w^-$ and 92% for *Mesh1 LOF* (Fig. 3a). On the starvation medium, the hatching date of *Mesh1 LOF* and $w^-$ larvae were around the fourth and the second day, respectively, after transfer. This result is consistent with a previous study that reported that *Mesh1 LOF* larvae showed retarded growth[25]. Despite the delay of hatching of *Mesh1 LOF*, the hatching rate from larvae to pupae of *Mesh1 LOF* (75%) was the same as that of $w^-$ (74%) (Fig. 3a). To investigate the physiological importance of ppGpp for the starvation response, we measured the amount of ppGpp in *Drosophila* under rich and starved conditions. We used late third instar larvae (shown in Fig. 2c) as the control and larvae were grown in rich medium for 4 days after egg-laying (AEL), and

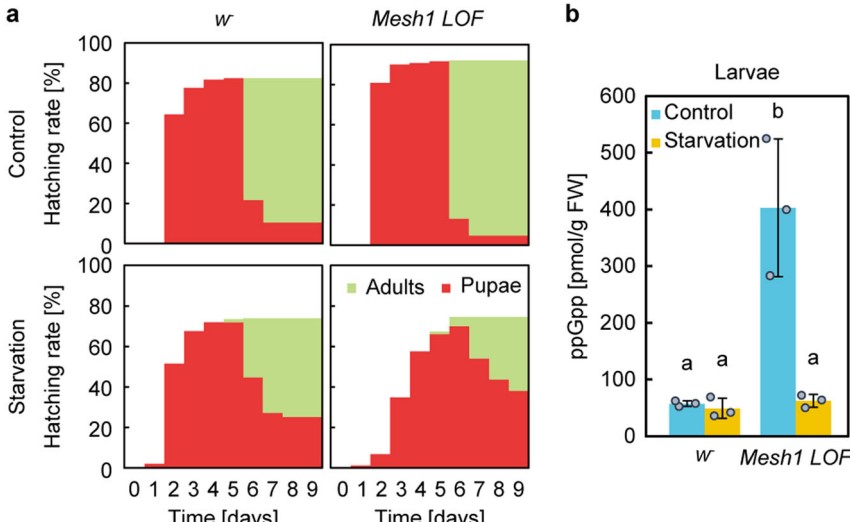

**Fig. 3 Mesh1-dependent control of ppGpp concentration is important for growth and starvation response. a** Hatching rate of the larvae (4 days after egg-laying) of $w^-$ and *Mesh1 LOF*, which were initially grown on standard medium and then transferred to normal (Control) or starvation conditions. The colors indicate the number of flies from each stage (%); red: pupae, green: adults. **b** Concentrations of ppGpp in *Mesh1 LOF* under normal and starvation conditions. Values represent the mean ± SD ($n = 3$ biologically independent samples). Different letters (a and b) indicate significant differences between groups ($p < 0.05$; Tukey's test).

then transferred to the starvation medium and incubated for one day for comparison with late third instar larvae. The starved *Mesh1 LOF* larvae weighed significantly more than the $w^-$ larvae (Supplementary Fig. 4a). Interestingly, the concentration of ppGpp in starved $w^-$ larvae did not differ from that of larvae in the control medium, whereas the concentration of ppGpp in *Mesh1 LOF* larvae was lower, i.e., the same as $w^-$ upon starvation (Fig. 3b and Supplementary Fig. 4b, c), indicating that ppGpp levels in $w^-$ larvae were normal under starvation conditions, but increased in rich medium in *Mesh1 LOF*. On the other hand, GTP levels (per FW) in *Mesh1 LOF* remained lower than that in $w^-$ upon starvation, although the GTP levels in $w^-$ were increased upon starvation (Supplementary Fig. 4d). These results suggested that the pupation delay of *Mesh1 LOF* (Fig. 3a) is not correlated with the ppGpp level itself, but is potentially due to the lowered GTP level in the mutant.

**ppGpp overaccumulation induces cell death in *Drosophila*.** To further investigate the function of ppGpp in *Drosophila*, we heterologously expressed the *Bacillus subtilis* SAS YjbM (Fig. 1a), which has strong ppGpp synthase activity[23]. We produced a *UAS-YjbM* line, which drives *yjbM* expression constitutively when it is crossed with the actin-specific driver line *Act5c-GAL4*. We found that offspring having both *UAS-yjbM* and *Act5c-GAL4* constructs were lethal, and progeny could not be produced. As an alternative, we transiently expressed *yjbM* under the control of the heat-shock-driven FLPase system (*hs-yjbM*) at day 3 AEL, when larvae show strong resistance against environmental changes[38]. *hs-yjbM* larvae could survive for several days after the induction of *yjbM*. The ppGpp chromatogram of the *hs-yjbM* exhibited a strong peak at 6.4 min (Fig. 4a), which corresponds to the retention time of the ppGpp standard (Fig. 1b), and indicated ~1200-fold higher ppGpp than in the control (Fig. 4b). After the heat shock, all the *hs-yjbM* died during the larvae and pupae stages (Fig. 4c), indicating that hyper-accumulation of ppGpp induces lethality in *Drosophila*, similar to that observed in the *E. coli spoT* mutant[39].

As systemic expression of *yjbM* is lethal in *Drosophila*, we could not further investigate the influence of ppGpp accumulation. Thus, we conducted site-specific expression of *yjbM* by

crossing with the eye-specific driver line *Glass multimer reporter (GMR)-GAL4* (*GMR-YjbM*). We also used the temperature-dependent GAL4 repressor GAL80ts to reduce leaky expression of YjbM. Before induction of *yjbM* expression, the surface of the eyes was smooth, but abnormally glossy (Fig. 5a), suggesting partial induction of cell death perhaps owing to leaky expression of *yjbM*. After induction of *yjbM*, the centers of the eyes became more yellow, supporting the cell-death hypothesis. Microscopic analysis of the optic nerves inside the eyes revealed fragmentation of the axons (Fig. 5b). Similar axon fragmentation has been reported in axon degeneration and neural cell death[40], suggesting that excess ppGpp is cytotoxic. On the other hand, the eye morphology of *Mesh1 LOF* did not differ from that of $w^-$, as seen in both optical images and confocal images of dissected eyes (Supplementary Fig. 5), suggesting that the cell-death phenotype observed in the *yjbM* OE line may not be achieved with physiological concentrations of ppGpp.

**Metabolic changes induced by altered Mesh1 expression.** We next performed metabolome analysis of *Mesh1 LOF* and *Mesh1 GOF* to investigate Mesh1 function on metabolism. The levels of certain metabolites were significantly altered in the mutants compared with controls (Fig. 6, Supplementary Fig. 6, and Supplementary Tables 2 and 3). In *Mesh1 LOF*, tricarboxylic acid (TCA) cycle metabolites including 4-hydroxybenzoate, fumarate, and malate were higher than in WT (CS). In *Mesh1 LOF*, the level of sedoheptulose 7-phosphate (S7P), an intermediate in the pentose phosphate pathway, was lower than that in WT (Supplementary Table 2). Intermediates in the transsulfuration pathway, CySSG was higher, and taurine and NADP+ levels were lower in *Mesh1 LOF* than in the WT (Fig. 6a and Supplementary Table 2). In *Mesh1 GOF*, two arginine metabolites, citrulline and creatine, were lower than in the control (Fig. 6b and Supplementary Table 3). Some intermediates in the methionine cycle and the transsulfuration pathway were also altered in *Mesh1 GOF*. Specifically, methionine and cystathionine levels were lower, and adenosine and pyroglutamic acid levels were higher in *Mesh1 GOF* than in the control (Fig. 6b and Supplementary Table 3). The levels of six metabolites were altered in *Mesh1 LOF* and *Mesh1 GOF*, and three of them, including the two TCA cycle

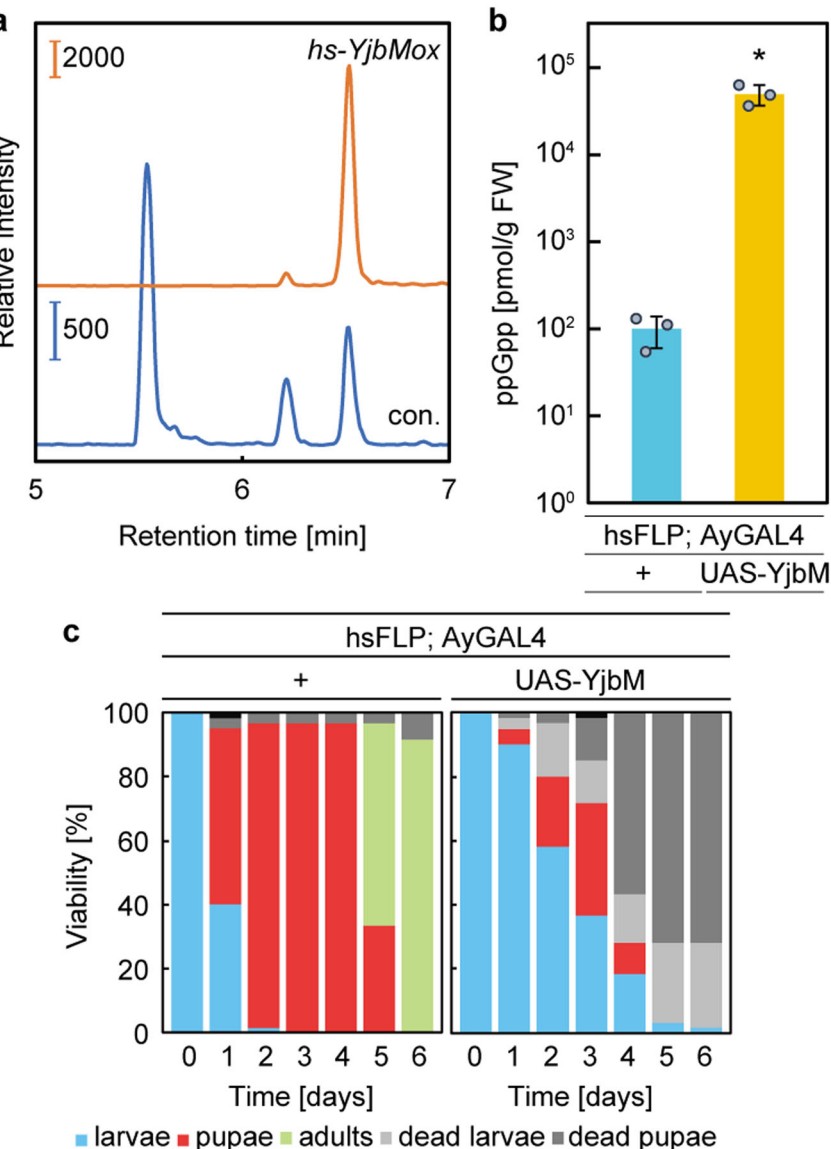

**Fig. 4 Growth arrest due to excess ppGpp. a** Guanosine tetraphosphate-specific MS chromatograms of nucleotides extracted from *hs-yjbM OE* (a heat-inducible *yjbM* overexpression line) and control larvae. **b** The concentration of ppGpp extracted from *hs-yjbM OE* larvae and the control (1 day after induction). Values represent the mean ± SD (n = 3 biologically independent samples). *p = 0.0227; two-sided Student's *t*-test. **c** *Drosophila* development after induction of ppGpp production. Values represent the mean (n = 3 independent experiments). The colors indicate each stage of flies; blue: larvae, red: pupae, green: adults, light gray: dead larvae, gray: dead pupae.

metabolites, malate and fumarate, were higher in *Mesh1 LOF* and lower in *Mesh1 GOF* (Fig. 6 and Supplementary Fig. 6). These results imply that Mesh1 influences both the TCA cycle and urea cycle in mitochondria, as well as the transsulfuration pathway and pentose phosphate pathway in the cytosol. In *Mesh1 GOF*, carboxymethyl lysine and pyroglutamic acid, one of the advanced glycoxidation end products and a glutathione metabolite, respectively, were increased (Fig. 6b and Supplementary Table 3). Advanced glycoxidation end products are involved in aging and in the development of many degenerative diseases in humans[41]. These results suggested that Mesh1 is important for maintaining some metabolic homeostasis in animal cells[42].

## Discussion

It is well established that ppGpp is a bacterial and plastidial second messenger[1–3]. However, ppGpp function in metazoa has not been determined because ppGpp had not been detected in

metazoan cells in spite of many trials[27]. Here we developed a high-sensitivity quantification method for ppGpp in metazoan cells by modification of our previous ppGpp quantification method used for plants and bacteria[32]. In the method, total nucleotide pools extracted with 2 M formic acid are subjected to a reversed-phase/weak anion exchange column in buffer containing ethylenediaminetetraacetic acid (EDTA), by which low amounts of ppGpp extracted from metazoa were efficiently concentrated. It is possible that chelating excess cations by EDTA in the solvent helps ppGpp adsorb to the anion exchange column. A chloroform extraction step was also added before LC-MS/MS, which resulted in enhancement of the ppGpp peak. The established method allowed us to stably detect small amounts of ppGpp and the unknown guanosine tetraphosphate molecule from *Drosophila* and human cells by LC-MS/MS (Fig. 1b and Supplementary Figs. 1 and 2). Given fragmentation patterns of ppGpp and the unknown molecule are nearly identical (Supplementary Fig. 1b), the structures of the two must be very similar. One possible

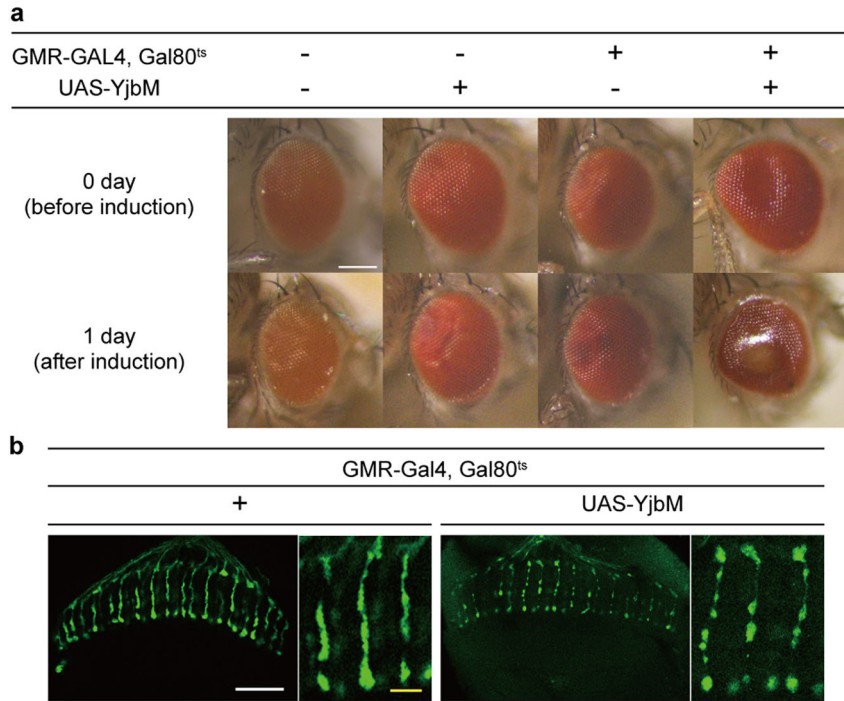

**Fig. 5 Excess ppGpp promotes cell death. a** Optical images of eyes (bar = 100 μm) and **b** confocal fluorescence images of axons (visualized with red fluorescent protein; white bar = 20 μm, yellow bar = 5 μm) of flies overexpressing *yjbM* by the eye-specific *GMR* promoter. Flies incubated at 18 °C 1 day after emergence (before induction; upper) were transferred to 31 °C, to induce overexpression of *yjbM* in the eye. Images were acquired 1 day after induction.

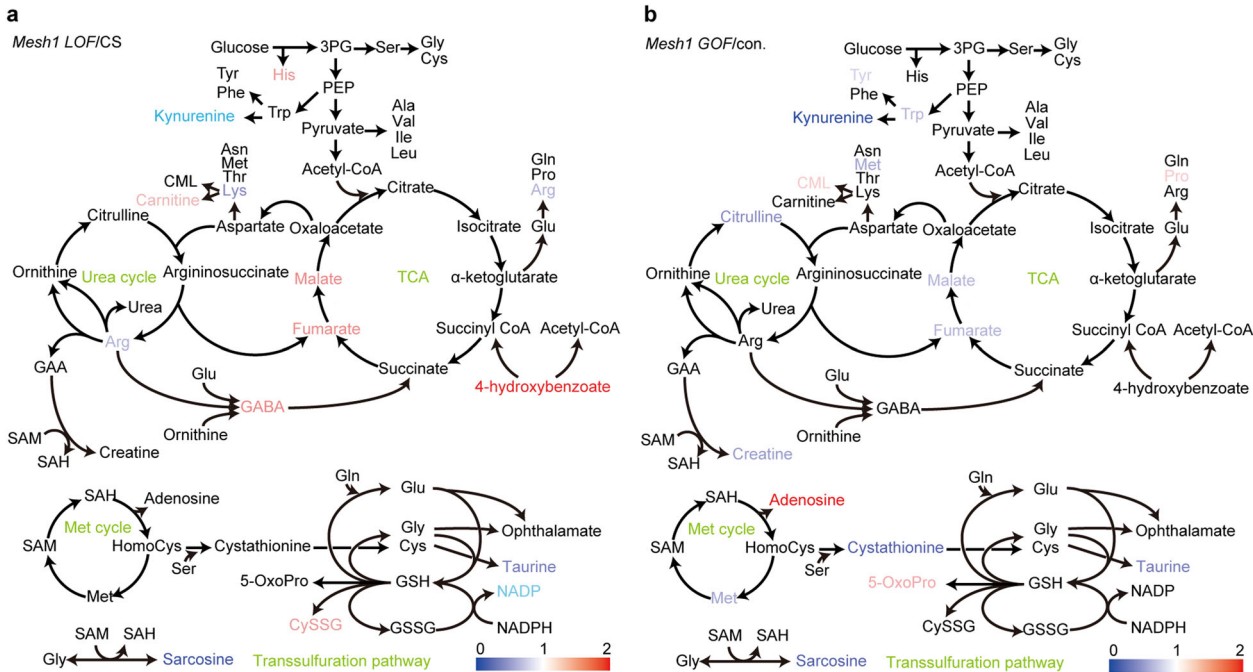

**Fig. 6 Metabolite changes in *Mesh1 LOF* and *Mesh1 GOF*.** Statistically significant changes in the concentrations of metabolites in *Mesh1 LOF* larvae compared with concentrations in *w⁻* larvae (*p* < 0.05, *n* = 6 biologically independent samples; two-sided Student's *t*-test) (**a**), and in *Mesh1 GOF* larvae compared with those in the control (*p* < 0.05, *n* = 6 biologically independent samples; two-sided Student's *t*-test) (**b**). Fold-change values in metabolite accumulation are expressed along a color gradient. 3PG: 3-phosphoglyceric acid; 5-OxoPro: pyroglutamic acid; CML: carboxymethyl lysine; CysSG: cysteine-glutathione disulfide; GAA: guanidinoacetic acid; GABA: gamma-aminobutyric acid, GSH: glutathione; GSSG: glutathione disulfide; PEP: phosphoenolpyruvate; SAH: *S*-adenosylhomocysteine; SAM: *S*-adenosylmethionine.

candidate for the unknown molecule is guanosine 2′,5′-bis(pyrophosphate), although this molecule has never been detected in cells. Further structural characterization of the unknown molecule should be performed in the future.

By use of the established method, we succeeded in detecting ppGpp in metazoan cells (Fig. 2 and Supplementary Figs. 2 and 3). ppGpp levels in Drosophila pupae were significantly higher than those in other stages (Fig. 2a and Supplementary Fig. 3a), implying that ppGpp is involved in metamorphosis. Mesh1 LOF showed higher levels of ppGpp (Fig. 2c and Supplementary Fig. 3c), suggesting that Mesh1 works as a ppGpp hydrolase in vivo in Drosophila, and that ppGpp metabolic systems have been conserved in metazoa. One could expect that ppGpp would be accumulated upon starvation in Metazoa as it is in bacteria[1]; however, ppGpp levels in $w^-$ before and after starvation treatment were the same (Fig. 3b and Supplementary Fig. 4b), suggesting that ppGpp synthesis is not upregulated upon starvation in Drosophila. Alternatively, ppGpp synthesis may be upregulated only in small areas of specific organs under starvation conditions. Notably, the ppGpp levels in Mesh LOF were significantly reduced upon starvation (Fig. 3b), suggesting the existence of a ppGpp-specific hydrolase other than Mesh1. In Arabidopsis, ppGpp is rapidly accumulated upon light-to-dark transition and the accumulated ppGpp is degraded in several hours in the dark[32,43]. The efficient degradation of ppGpp in plants is achieved by not only RSHs, but also several nucleotide diphosphate linked some moiety X (Nudix) hydrolase enzymes, similar to the process that occurs in bacteria[44,45]. Given that Nudix enzymes are conserved in Metazoa[46], ppGpp levels in Metazoa are also likely controlled by multiple enzymes including Mesh1 and Nudix hydrolases, as well as unknown ppGpp synthases.

Although the physiological role of Mesh1-dependent ppGpp control in Metazoa is still largely unknown, our metabolome analyses suggested that enzymes involved in purine metabolism[47] are regulated by Mesh1 activity. In Mesh1 GOF, adenosine and adenine levels were higher and hypoxanthine level was lower than those in the control (Supplementary Table 3). In B. subtilis, (p)ppGpp directly inhibits certain enzymes involved in GTP synthesis, including guanylate kinase and hypoxanthine phosphoribosyltransferase[3], and this inhibition reduces the cellular GTP pool and retards transcription by reducing the concentrations of factors (e.g., ribonucleotides) necessary for transcription[48]. The reduced levels of hypoxanthine observed in Mesh1 GOF (Supplementary Table 3) were likely due to the reduction of ppGpp (Fig. 2d) and metazoan hypoxanthine phosphoribosyltransferase conserves the amino acid residues necessary for ppGpp binding in bacterial enzymes[49]. Together with the increased concentration of GTP in the Mesh1 LOF (Supplementary Fig. 3d), these results imply that Mesh1-dependent ppGpp control in metazoans regulates the GTP/ATP ratio by controlling purine nucleotide metabolism, as in bacteria. Moreover, other GTP-binding proteins could be ppGpp targets[50].

hMESH1 was recently found to have NADPH phosphatase activity[31]. The NADP(H) concentration in the hMESH1-knockdown line was ~2 times higher than that in the control[31], suggesting that Mesh1 has a dual function to sustain levels of intracellular ppGpp and NADPH. However, NADP+ concentrations in the Drosophila Mesh1 LOF were about half of that in WT ($P = 0.050$) (Fig. 6a and Supplementary Table 2); this result is the opposite of what is observed in the hMESH1-knockdown line. Although the reason for this variation is not known, one possibility is that null mutation of Mesh1 induces a decrease of the NADP(H) pool size to compensate for the constitutive increase of NADP(H) levels upon loss of NADPH phosphatase activity of Mesh1.

Recently, the impact of artificial accumulation of ppGpp in the yeast Saccharomyces cerevisiae was examined by heterologous

expression of E. coli relA, which revealed that accumulated ppGpp induces expression of some mitochondrial proteins such as cytochrome oxidase and the succinate dehydrogenase homolog SHH4, the latter of which catalyzes fumarate synthesis from succinate in the TCA cycle[51]. In this study, we found that Mesh1 LOF accumulates fumarate and malate (Fig. 6a and Supplementary Table 2), suggesting that highly accumulated ppGpp in Mesh1 LOF influences TCA cycle metabolisms in mitochondria. Previous microarray analysis of the Drosophila Mesh1 LOF indicated that expression of genes for mitochondrial function, lipid and carbohydrate metabolisms were altered in the mutant[25], supporting the hypothesis. The ppGpp-accumulating S. cerevisiae also has upregulated expression of 6-phosphogluconolactonase SOL4, transketolase TKL2, and transaldolase NQM1, all of which function in the pentose phosphate pathway[51]. TKL2 catalyzes the synthesis of S7P and glyceraldehyde 3-phosphate (GAP), and NQM1 catalyzes synthesis of fructose 6-phosphate and erythrose 4-phosphate from S7P and GAP[52]. The concentration of S7P in Mesh1 LOF was significantly decreased compared to that in the control (Supplementary Table 2), suggesting that accumulated ppGpp changes metabolism in the pentose phosphate pathway in Drosophila. Given the pentose phosphate pathway requires NADP+ [53], one could assume that inhibition of ppGpp hydrolase and NADP(H) phosphatase activities of Mesh1 result in increasing ppGpp and NADP(H) pool sizes that could consequently upregulate pentose phosphate pathway activity. This further supports the physiological bases of the dual activities of Mesh1 as ppGpp hydrolase and NADPH phosphatase to control specific metabolisms in metazoa.

We observed retarded growth of Mesh1 LOF under starvation conditions (Fig. 3a), as reported previously[25]. The pupation rate of Mesh1 LOF was the same as that of WT, suggesting that the viabilities of Mesh1 LOF and WT were comparable (Fig. 3a). On the other hand, the viability of Mesh1 LOF was reported to be 70% of that in WT under starvation conditions in the previous study[25]. This inconsistency may be due to different timing of transfer of the larvae to the starved medium. In the previous study, larvae were transferred to starvation medium at 40 h AEL, whereas we transferred larvae to the starvation medium at 4 days AEL. These results suggest that Mesh1 function in Drosophila is more important to endure the starvation at early larval stages than in later periods. It should be noted that it is still not possible to distinguish phenotypes caused by loss of ppGpp hydrolase and/or NADPH phosphatase activities in Mesh1 LOF and GOF. Clearly, further studies are needed to determine how ppGpp works in metazoan cells.

Overall, our method for quantifying ppGpp in animal cells should provide opportunities for investigating the stringent response in metazoa. Since our research revealed that ppGpp and its metabolic system are conserved in metazoan cells, which was a decade-old enigma, this result should provide useful insights for future studies such as understanding how ppGpp is synthesized and how it functions in metazoan cells.

## Methods

**Fly strains, genetics, and growth conditions.** Flies were kept in standard Drosophila medium at 25 °C, except for activating GAL80ts (18 °C). To activate the heat-shock (hs) promoter, flies were transferred from 18 °C to 31 °C 1 day after emergence. To calculate the concentration of ppGpp in each stage of Drosophila, eggs were collected for 19 h on an apple juice medium, larvae at third instar larvae, pupae 48 h after pupation, virgin during 0–8 h after hatching, and adults 4 days after culture on the standard medium. To induce yjbM OE on day 3 AEL, hs-yjbM was heat-shocked for 30 min at 37 °C and then maintained for 10 min at room temperature. This heat-shock procedure was then repeated. To compare the pupation rate and ppGpp level under starvation conditions, eggs from flies of each genotype were collected over 2 h on an apple juice medium. Eggs were aged on standard Drosophila medium at room temperature (25 °C) and the standard medium was used as the normal growth condition (see below for recipe). To assess

the effects of starvation, larvae at 4 days AEL under medium were transferred to starvation medium (see below for recipe). The following fly stocks and mutant alleles were used: CS, $w^{1118}$ ($w^-$), Mesh1 LOF mutant, UAS-Mesh1, UAS-yjbM, Act5C-Gal4, Act5c-Gal4-tubGAL80$^{ts}$, hsFLP-AyGAL4-UAS-GFP, and GMR-myr-RFP-GMR-GAL4-tubGal80$^{ts}$. The UAS-yjbM (residues 1–636 with a C-terminal FLAG tag) fly was produced as follows. At first, B. subtilis yjbM was amplified with a primer pair, 5′-AGGGAATTGGGAATTCAAAATGGATGACAAACAGTGG-3′ and 5′-ATCTGTTAACGAATTCTTACTTATCATCATCATCCTTATAATC-3′, and synthesized yjbM gene[54] as a template. The amplified fragment was cloned into EcoRI-cut pUASTattB[55] by In-fusion cloning kit (Clontech), which was integrated into the specific site (VK31) of chromosome III with the $\Phi$C31 system[55]. Transformants were produced by BestGene, Inc. (USA). Mesh1 LOF and UAS-Mesh1 were kindly provided by J. Chung (Seoul National University)[25]. CS was used as the WT strain for the detection of ppGpp from eggs. $w^-$ was used as the control strain for comparison with mutants. Eye-specific expression was achieved using the GAL4-UAS expression system[37]. GMR-GAL4 was used to induce expression in eyes[56]. To analyze the yjbM OE line, we made +/+; +/+; Act5c-GAL4, Tub-Gal80$^{ts}$/UAS-yjbM (or +) (yjbM OE), hs-FLP/w; AyGAL4, UAS-GFP/+; +/UAS-yjbM (or +) (hs-yjbM OE), and GMR-myr-RFP/w; GMR-GAL4, tub-Gal80$^{ts}$/+; +/UAS-yjbM (or +) (GMR-yjbM OE). To analyze the Mesh1 GOF, we crossed UAS-Mesh1 flies with yw; Act5c-Gal4/TM6B and $w^-$ flies with yw; Act5c-Gal4/TM6B as a control. We sampled +/yw; UAS-Mesh1/+; +/ Act5c-Gal4 as the Mesh1 GOF line and w/yw; +/+; +/ Act5c-Gal4 as the control.

**Gene expression**. To evaluate the effects of ppGpp accumulation, we used the GAL4/UAS system[37]. We used transformants that had UAS-yjbM or UAS-Mesh1, which were silent in the absence of GAL4. They were crossed with flies expressing GAL4. In the progeny of this cross, GAL4 protein activated transcription of yjbM or Mesh1 by binding to a UAS[37]. GMR-GAL4 was used for induction of photoreceptor gene expression. The GMR sequence is an engineered transcription regulatory region containing a binding site for the protein gl, a photoreceptor-specific transcription factor[56]. myr is the sequence from D. melanogaster Src64B that includes the myristoylation signal and is used to target a protein to the plasma membrane. To induce yjbM expression at a specific developmental time, we used GAL80$^{ts}$ or the flippase (flp)/FRT system in conjunction with the GAL4/UAS system, which was controlled by the hs promoter[38,57]. GAL80$^{ts}$ is a temperature-sensitive GAL80$^{ts}$ gene. At 18 °C, GAL80$^{ts}$ binds GAL4 and this heterodimer works as a GAL4 repressor. Upon transfer to a temperature above 29 °C, GAL80$^{ts}$ is unable to bind GAL4 and its repressive function is lost[57]. Thus, from routine culture at 18 °C, flies were transferred to 31 °C when induction of yjbM was desired. With the flp/FRT system, the transformants had three genes to control gene expression: heat shock-flp fusion gene, the Act5C promoter-GAL4 fusion gene with a FRT cassette containing a transcription termination signal, and UAS-yjbM. Before heat shock, Act5C-GAL4 was interrupted by the transcription termination signal in the FRT cassette. Heat shock induced the expression of flp encoding the flippase, which excised the FRT cassette from the fusion gene. The GAL4 protein expressed from the rearranged fusion gene activated transcription of UAS-yjbM[38].

**Culture medium for Drosophila**. For the standard medium, 15 g soybean flour, 10 g agar, 100 g cornmeal, 30 g yeast, 30 g malt, and 50 g glucose were mixed in 1 l water and then heated at 90 °C for 1.5 h. After the mixture had cooled, 2 ml propionic acid and 2.5 g nipagin in 5 ml 100% ethanol were added. The mixture was then dispensed into columned vials or bottles until it covered the bottom to a depth of ~2 cm. The starvation medium consisted of 20% (w/v) sucrose-agar medium containing 0.5% ethanol and 0.2% propionic acid in distilled/deionized water[58].

**Germ-free larvae**. The $w^{1118}$ flies were obtained from the Bloomington Stock Center (Bloomington, IN) and used as the standard strain. Adult flies were maintained in autoclaved standard yeast medium containing tetracycline (50 μg/ml) for three generations to eliminate the common intracellular bacteria Wolbachia pipientis. Embryos of the flies were then bleached using 2.7% sodium hypochlorite and washed twice in 70% ethanol, followed by three washes in sterile water. Washed embryos were transferred to autoclaved yeast vials without tetracycline. The third instar larvae were collected and bacterial contamination in larvae was assessed by PCR using 16S rDNA universal primers 8FE (5′-AGAGTTTGAT CMTGGCTCAG-3′) and 1492 R (5′-GGMTAGCTTGTTACGACTT-3′).

**Cell culture**. Human cervical cancer cells (HeLa) and human embryonic kidney cells (PEAKrapid) were obtained from the American Type Culture Collection (Manassas, VA). Cells were cultured in Dulbecco's modified Eagle medium (Sigma-Aldrich, St. Louis, MO) supplemented with 10% fetal bovine serum (Thermo Fisher Scientific, Waltham, MA), 100 U ml$^{-1}$ penicillin (Nacalai Tesque, Kyoto, Japan), and 100 μg ml$^{-1}$ streptomycin (Nacalai Tesque) at 37 °C in 5% CO$_2$. Drosophila S2 cells (GIBCO, R690-07) were cultured at 25 °C in Schneider's medium supplemented with 10% heat-inactivated fetal bovine serum (Thermo Fisher Scientific) and 1× antibiotic–antimycotic (Thermo Fisher Scientific). Plastic Petri dishes were used for cell culture, and harvested cells were frozen at −70 °C until use.

**Phenotypic analysis of Drosophila**. To analyze the effect of ppGpp on cell proliferation under starvation conditions, 50 flies of $w^-$ and those of Mesh1 LOF that had developed for 3 days (i.e., 3 days AEL) were transferred and kept on standard or starvation medium. The number of pupae and adults in the medium were counted every 24 h. hs-yjbM OE was used for the analysis of the effect of excess ppGpp. To induce yjbM OE at day 3 AEL, hs-yjbM OE was heat-shocked for 30 min at 37 °C and then held at 10 min at room temperature, and this heat-shock procedure was then repeated. After heat shock, the larvae were cultured for 24 h on standard medium containing Formula 4–24® Instant Drosophila Medium, Blue (Carolina Biological Supply Company) at 25 °C. Larvae for the two genotypes were then gathered after incubation: hs-FLP-GFP/w; AyGAL4, UAS-GFP/+, +/yjbM (or +). Twenty larvae were transferred to standard medium; every 24 h, all flies were assessed as living or dead and as larvae, pupae, or adults. To observe the effects of excess ppGpp on eye development, GMR-yjbM adults and other control genotypes were transferred from 18 °C to 31 °C 1 day after emergence. After 0 (as a control) or 1 day incubation, the eyes and dissected eyes were examined microscopically.

**Immunohistochemistry and imaging**. Brains of the 1-day-old adult flies (or those of 1 day after induction of YjbM) was dissected in phosphate-buffered saline and fixed by 4% paraformaldehyde solution. After several washes, the brains were incubated with primary (o/n) and secondary (2 h) antibodies sequentially. After immersing in the Vectashield mounting medium (Vector Laboratories Inc.), the brains were mounted vertically on the slide glass[59]. The primary antibody for the photoreceptor cell-specific membrane protein, chaoptin, was mAb24B10 (1:50, DSHB). The secondary antibodies were conjugated to Alexa488 (1:400, Life Technologies). Images were obtained with Nikon C2+ confocal microscope and processed with NIS-elements AR and Adobe Photoshop.

**Quantification of ppGpp and GTP by LC-ESI-MS/MS**. ppGpp was extracted from cells and quantified as described by Ihara et al.[32] with slight modifications as detailed below. Standards for ppGpp (≥85%), GTP (92%), and GTP (>99%) were purchased from Trilink, TaKaRa, and Bio Basic, Inc., respectively, and used without further purification. Fifty flies grown under normal conditions or ~65 mg of flies grown under starvation conditions were collected for all quantifications, except that only one larva was collected for quantification of ppGpp in YjbM OE. The collected samples were frozen in liquid nitrogen, broken into pieces using a pestle and mortar, and then extracted in 3 ml of 2 M formic acid/1 mM EDTA pH 4.5. Extracts were incubated for 30 min on ice. Subsequently, 50 mM ammonium acetate/1 mM EDTA pH 4.5 was added to the crushed tissue to a volume of 6 ml and the extracts were divided into two 3 ml portions. To estimate the ppGpp recovery rate[32], 25 μl of 2 μM ppGpp was added to one of the two solutions (the standard mixed sample). For GTP quantification, this standard mixing step was omitted, perhaps because the GTP recovery rate was constant, perhaps because of its high concentration. To remove hydrophobic debris, 1.5 ml chloroform was added, the mixtures shaken three times, centrifuged at 1600 × g for 1 min at room temperature, and each aqueous (upper) layer transferred to a new tube.

The samples were then subjected to solid-phase extraction. A 1-ml OASIS WAX column 30 mg (Waters) was rinsed with 1 ml of methanol followed by 1 ml ammonium acetate pH 4.5. Each sample solution was loaded onto the column, which was then washed with 1 ml of ammonium acetate pH 4.5 followed by 1 ml of methanol. Nucleic acids were eluted from the column with 1 ml methanol/deionized H$_2$O/25% aqueous ammonia (20:70:10) solution. The eluate was lyophilized in a Proteosave SS 15 ml Conicaltube (Sumitomo Bakelite), dissolved in 200 μl in Milli-Q water, and then filtered through a FavorPrep$^{TM}$ Plasmid DNA Extraction Column (FAVORGEN). A 94 μl portion of the eluate was then mixed with 6 μl acetonitrile in a PSVial (AMR).

A 10 μl aliquot of this solution was injected into an Acquity UPLC system (Waters) and separated on an ACQUITY UPLC® BEH C18 column (2.1 × 50 mm, 1.7 μm particle size; Waters). Nucleotides were resolved with a linear gradient of two mobile phases: solvent A (500 ml of 8 mM N,N-Dimethylhexylamine with 80 μl of acetic acid) and B (acetonitrile) at a flow rate of 0.3 ml min$^{-1}$, as follows: initially 0% B; 0–40% B over 10 min; 40–0% B over 10.5 min; reequilibration at 0% B over 20 min. Effluents from UPLC were introduced online into an ACQUITY TQD tandem quadruple mass spectrometer with an electrospray ionization (ESI) interface (Waters). Argon was used as collision gas at 0.3 ml min$^{-1}$. Negative ESI-MS/MS detection was carried out in the multiple-reaction monitoring mode or product ion scan mode using the following parameters: capillary voltage, 2.5 kV; source temperature, 150 °C; desolvation temperature, 400 °C; cone gas flow, 50 l h$^{-1}$; desolvation flow, 800 l h$^{-1}$; LM1 resolution, 13; LM 2 resolution, 15. The cone voltage and collision energy were 44 V and 44 eV, respectively, for ppGpp, and 40 V and 32 eV, respectively, for GTP. The multiple-reaction monitoring transitions were 602→159 for ppGpp and 522→159 for GTP.

The ESI source was set to negative-ion mode with a capillary voltage of 2.5 kV. The source and desolvation temperatures were set to 150 °C and 400 °C, respectively. ppGpp was detected in multiple-reaction monitoring mode. The cone voltage and collision energy were 44 V and 44 eV, respectively. The mass of the parent ion (602 m/z) was selected by the first quadrupole and fragmented in the collision cell to the target ion (159 m/z).

ppGpp and GTP levels were quantified against standard curves. Standard solutions consisted of 20 μl of 2 μM ppGpp or 0.4 mM GTP mixed with 12 μl of

acetonitrile and 168 μl of Milli-Q water in a PSVial (AMR). The standard curves were linear at concentrations between 1.5625 nM and 200 nM for ppGpp, and 2500 nM and 40,000 nM for GTP. The calibration curves were used when the $r^2$ values of the plotted calibration data were >0.99. The final ppGpp concentration in each sample was corrected based on its recovery rate calculated by the ppGpp standard mixed sample, as reported[32]. Obtained data are shown in Supplementary Data File S1.

The LC-MS/MS-based ppGpp quantification method described above was used to detect pppGpp with the multiple-reaction monitoring transitions at 682→159 and 234. For the experiment, pppGpp standard[12] was prepared as follows. At first, B. subtilis yjbM was amplified with a primer pair, 5′-AAGGAGATATACATAT GGATGACAAACAGTGGG-3′ and 5′-GGTGGTGGTGCTCGAGTTGCTGCTCA GATCCTTTC-3′, and synthesized yjbM gene[54] as a template. The amplified fragment was cloned into NdeI-XhoI-cut pET29a (Merck) by In-fusion cloning kit (Clontech). The C-terminal His-tagged YjbM was expressed in the E. coli strain BL21(DE3) and purified by His-bind resin (Novagen) according to the manufacturer's recommendations. The purified YjbM was mixed with 5 mM GTP and 5 mM ATP in a buffer containing 50 mM Tris/HCl pH 7.5, 200 mM NaCl, 20 mM MgCl$_2$, and 20 mM KCl, and incubated for 30 min at room temperature, followed by chloroform extraction. The aqueous phase that contained the nucleotides was subjected to anion exchange chromatography (HitrapQ HP, 1 mL; GE Healthcare). A gradient ranging from 0–2 M NaCl was used to separately elute the nucleotides.

**Metabolome analysis by CE-TOF MS**. Ten flies of late third instar larvae were collected for metabolome analysis. Each sample was extracted in 500 μl methanol containing 8 μM of each of two reference compounds, namely methionine sulfone for the cation analysis and camphor 10-sulfonic acid for the anion analysis, using a Retsch mixer mill MM310 at a frequency of 27 Hz for 1 min. The extracts were then centrifuged at 15,000 × g for 3 min at 4 °C. The supernatant was transferred into a tube, and 500 μl chloroform and 200 μl water were added to the tube to perform liquid–liquid extraction. The upper layer (water) was transferred to a new tube, and evaporated for 30 min at 45 °C by a centrifugal concentrator to obtain two layers. The resulting partially evaporated liquid was then centrifugally filtered through a PALL Nanosep 3-kDa cutoff filter at 9100 × g for 90 min at 4 °C to remove high-molecular-weight compounds such as oligo-sugars. The filtrate was evaporated to dryness for 120 min using a centrifugal concentrator. The residue (ca. 25 mg for each sample) was dissolved in 20 μl water containing 200 μM of each of the internal standards, namely 3-aminopyrrolidine for the cation analysis and trimesic acid for the anion analysis, that were used to compensate for differences in migration time in the peak annotation step.

All CE-TOF MS experiments were performed using a Model G7100A CE Instrument (Agilent Technologies, Sacramento, CA), an Agilent G6224A TOF LC/ MS system, an Agilent 1200 Infinity series G1311C Quad Pump VL, an G1603A Agilent CE-MS adapter, and a G1607A Agilent CE-ESI-MS sprayer kit. The G1601BA 3D-CE ChemStation software for CE and G3335-64002 MH Workstation were used. Separations were carried out using a fused silica capillary (50 μm i.d. × 100 cm total length) filled with 1 M formic acid for cation analyses or with 20 mM ammonium formate pH 10.0 for anion analyses as the electrolyte. The capillary temperature was maintained at 20 °C. The 15 nl sample solutions were injected at 50 mbar for 15 s. The sample tray was cooled below 10 °C. Prior to each run, the capillary was flushed with electrolyte for 5 min. The applied voltage for separation was set at 30 kV. Methanol/water (50% v/v) containing 0.5 μM reserpine was delivered as the sheath liquid at 10 μl min$^{-1}$. ESI-TOF MS was conducted in the positive-ion mode for cation analyses or in the negative-ion mode for anion analyses, and the capillary voltage was set at 30 kV. A flow rate of heated dry nitrogen gas (heater temperature 300 °C) was maintained at 10 l min$^{-1}$. The fragmentor, skimmer, and Oct RFV voltages were automatically set at optimum values. Automatic recalibration of each acquired spectrum was performed using the masses of reference standards. The methanol dimer ion ($[2M + H]^+$, m/z 65.0597) and reserpine ($[M + H]^+$, m/z 609.2806) for cation analyses or the formic acid dimer ion ($[2M-H]^-$, m/z 91.0037) and reserpine ($[M-H]^-$, m/z 607.2661) for anion analyses provided the lock mass for exact mass measurements. Exact mass data were acquired at a rate of 1.5 cycles s$^{-1}$ over a 50–1000 m/z range. In every single sequence analysis (maximum 36 samples) with our CE-TOF MS system, we analyzed the standard compound mixture both before and after each analysis of a sample. The detected peak area of the standard compound mixture was compared at regular intervals to assess sensitivity and reproducibility. The standard compound mixture was composed of major detectable metabolites, including amino acids and organic acids, and this mixture was freshly prepared at least once every 6 months. For all analyses, there were no differences in the sensitivity of detection of the standard compounds mixture.

An original data file (d) was converted to a unique binary file (.kiff) using in-house software. Peak picking and alignment were performed using another in-house software package in which peaks were picked and aligned among samples automatically. Using the detected m/z and migration time values of standard compounds, including internal standards, peaks were annotated automatically using the same software. For normalization, the individual areas of the detected peaks were divided by the peak area of the relevant internal reference standards. Using the calibration curves for standard compounds, peak area values were

converted into values corresponding to amounts. Obtained data are shown in Supplementary Data File 1.

**Statistics and reproducibility**. Statistical significance of data was tested by two-sided Student's t-test and/or Tukey's test by used of Excel and MEPHAS (http:// www.gen-info.osaka-u.ac.jp/MEPHAS/tukey-e.html), respectively. The sample size (n) and the nature of replicates have been given wherever relevant.

**Reporting summary**. Further information on research design is available in the Nature Research Reporting Summary linked to this article.

## Data availability

The dataset generated and analyzed in the current study is available as Supplementary Data File 1. All other data (if any) are available from the corresponding author upon request.

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

## Acknowledgements
We thank J. Chung (Seoul National University) for proving fly strains. We acknowledge support from JSPS KAKENHI Grant Number 19K22418 (to S.M.). D.I. is supported by a Grant-in-Aid for JSPS Fellows Grant Number 18J12607.

## Author contributions
D.I. and S.M. designed the research. D.I. and Y.I. quantified ppGpp and GTP. D.I., H.K., and T. Suzuki constructed and characterized *Drosophila* mutants. A.O. performed metabolome analysis. T. Shibata and S.-I.K. grew and cultivated germ-free flies. N.N. and T.A. maintained and sampled cultured cells. D.I. and S.M. wrote the manuscript. All authors contributed to the discussion of the results.

## Competing interests
The authors declare no competing interests.
