## [Peer Review File · Communications Biology]

Reviewers' comments:

Reviewer #1 (Remarks to the Author):

In this paper, the authors detect ppGpp in *Drosophila melanogaster* during different developmental stages. The use of germ-free *Drosophila* larvae and antibiotic treated cultured cells suggest that the ppGpp detected is not all from bacterial origin. Similar results were obtained with human cells. Varying the levels of Mesh-1 hydrolase by overexpression or deletion, produce the expected changes on the ppGpp levels, that are accompanied by metabolic changes. The authors show that high levels of ppGpp produce cellular lethality. Several attempts have been made to detect ppGpp in metazoan cells but without much success until now. Ito et al. seem to have succeeded on detecting ppGpp by LC-ESI-MS/MS in metazoa.

1. RSH enzymes usually are classified as long RSH enzymes, that contain domains for synthesis and hydrolysis (such as SpoT), and small alarmone synthetase (SAS, like Yjbm) or small alarmone hydrolases (SAH, like Mesh-1). For clarity, this could be mentioned in the introduction or figure 1A. Also, to unify nomenclature between different species, Yjbm is referred as RelQ by several authors.

2. In figure 1B and C, the authors detect a peak for GTP, ppGpp and then an unknown peak in between, present also in the ppGpp standard. Can it be pGpp? Some synthetases, like RelQ from *Enterococcus faecalis* produces pGpp (doi 10.1128/JB.00324-15). Also ppGpp and pppGpp can be degraded to pGpp in *B. subtilis* by a NuDiX enzyme (<https://doi.org/10.1101/2020.03.23.003749>). pGpp has a differential motility than GTP in TLC and LC-MS, being usually found between GTP and ppGpp, as the unknown compound. It could be determined by running a pGpp standard as done by ppGpp alone. Considering that the levels of the unknown peak changes between the differential stages of *Drosophila* (figure S2) and between different human cell cultures (figure 1C), seem relevant to determine if this unknown peak is pGpp or not.

3. Together with ppGpp, many synthetases produce also pppGpp, although it is not mentioned in the paper. Can pppGpp be detected with this method? Or is pppGpp not detected in *Drosophila*? This should be discussed in the paper as either deficiency of the method or as a differential characteristic.

4. The figure 2 seem to show divergences on the levels of ppGpp in pupae stage between panels A and D. In figure 2A the levels of ppGpp in pupae are up to 250 pmol/g FW while in figure 2D seems to be up to 600 pmol/g FW. Are the levels of ppGpp of Mesh1 *gof* from figure 2D being compared to *w-* (as indicated in the legend) or to *yw*; *Act5c-Gal4* (as suggested in the methods section)? If they are compared to *yw*, please, indicate it in the figure legend. Then, why are the levels of ppGpp in *yw* pupae higher than *w-*? Please, add the ppGpp measurements of *yw* pupae in the table S1 too.

5. The results shown in figure 3 seem a bit challenging and raised the following concerns:

a. The authors show hatching rates from larvae to pupae and from pupae to adult in figure 3A, but they talk about the final hatching rate from larvae to adult in the text. At the same time, they only show ppGpp levels on larvae (fig. 3B). From my point of view (with not much knowledge in *Drosophila*), wouldn't be better just focusing on the hatching rate from larvae to pupae? If not, were the ppGpp levels measured during pupae stage during starvation conditions? It seems to me that to discuss properly the effects of ppGpp on the hatching from larvae to adult, the levels of ppGpp should be determined in pupae stage too, not only in larvae as shown in fig. 3B.

b. The authors show that there is a decrease on the hatching rate in *wt* flies between control and starvation conditions (72% to 49%), but no differences in the ppGpp levels are detected. To me, these data suggest that ppGpp is not increased during the starvation conditions used in this study and that it doesn't seem responsible for the hatching rate decrease observed in *wt* flies.

c. The authors performed the same experiment with Mesh1 *lof*. In control conditions, the authors show an increase on the levels of ppGpp (7-fold) in Mesh1 *lof* and an increase on the hatching rate (72% to 87%). However, when they determine the levels of ppGpp in Mesh1 *lof* starved larvae,

they see low levels of ppGpp, similar to the ones observed in wt larvae. No explanation is given for this decrease. Also, they observe a decrease on the hatching rate compared to wt (48% to 37%) during starvation although having similar ppGpp levels. The authors suggest that the pupation delay of Mesh1 lof (during starvation) is due a decrease on the GTP levels (figure S3D). The results in figure S3D are performed in control conditions, where the lower levels of GTP are due the high levels of ppGpp, as suggested by the authors. Why the GTP levels of Mesh1 lof during control conditions are used to explain the challenging results observed during starvation conditions?

To me, the results in figure 3 seem to contradict the authors when they suggest that "control of ppGpp concentration is important for growth and the response to starvation". At the same time, the mesh1 lof results during starvation remain unexplained. I encourage the authors to clarify these results from figure 3. Alternatively, I would recommend eliminating the mesh1 lof results and only discuss the results observed in WT.

6. In figure 1C, the ppGpp levels of different culture cells and germ-free (GF) flies are determined. The authors show a chromatogram for GF and another for GF control. What is the difference between GF and GF control? Please, indicate it in the figure legend.

7. In figure 2A and S3A, blue bars represent males or mixed population. I understand that sex cannot be assessed during eggs or pupae stage, resulting in a mixed population. I assume that in the other stages blue bars refer to male population. Please, use different colors to differentiate between mixed populations and males.

8. In figures 3, S3 and S4 the statistical significance of the differences is indicated by using letters, but no other explanation is given. What does the different letters mean?

9. In Hesketh et al. 2017 (DOI: 10.1128/mBio.01047-17) the authors overexpressed RelA in *S. cerevisiae* to determine the effects of ppGpp over gene expression, and they determined that ppGpp upregulates transcription of genes encoding key mitochondrial proteins. It would be interesting to compare their results with the metabolic changes observed in *Drosophila*.

10. As mentioned by the authors in the discussion, Mesh1 seems to have a NADPH phosphatase activity. Do the authors see such effect in Mesh1 lof or Mesh1 gof? In table S2, there seems to be a 2-fold decrease on the levels of NADP in Mesh1 lof compared to WT. It would be interesting to discuss this on the paper.

Reviewer #2 (Remarks to the Author):

Review of Ito et al. *Com. Biol.* "ppGpp functions 1 as an alarmone in metazoan"

Ito et al propose that guanosine tetraphosphate (ppGpp) is present in measurable levels in *Drosophila melanogaster*, and that this ppGpp molecule would be hydrolyzed *in vivo* by the recently discovered RelA-SpoT homologue (RSH) Mesh1 hydrolase in certain conditions. They further propose that changes in ppGpp levels could be responsible for multiple Mesh1-mutant derived phenotypes and changes in the metabolome, including starvation phenotypes, reminiscent of ppGpp-dependent starvation phenotypes observed in bacteria and plants.

The main discovery of this study is the first direct observation of the ppGpp molecule in *Drosophila melanogaster* and more generally in metazoa. The authors observed ppGpp at various stages of development, under various growth conditions and also tested germ-free (GF) strains to insure that ppGpp was not produced by commensal bacteria. The authors also demonstrate that Mesh1 is necessary, at least in certain conditions, to maintain low ppGpp levels, hence providing insights into the role of Mesh1 as a ppGpp hydrolase *in vivo*. As for the diverse phenotypes observed in Mesh1 mutants, while they somewhat correlate with ppGpp levels in certain cases, it is not clear nor demonstrated whether the effect observed results from Mesh1 ppGpp hydrolase activity or its known NADPH phosphatase activity, or yet other functions yet to be discovered of this enzyme. Expression of a heterologous ppGpp synthase provokes a strong phenotype, along with a spike in

ppGpp levels, supporting the idea that high ppGpp levels are cytotoxic, but these effects are not necessarily directly related to the effects observed with Mesh1 mutants. Altogether, this manuscript is a valuable contribution to the literature. However, several passages require more detailed explanations and proper conclusions, while several conclusions about the effects of ppGpp in vivo require to be toned down, as for the most part no demonstration of the direct effect of ppGpp is provided.

Major comments

1. Line 68 and Fig. S1. How the authors explain a peak in GTP 92% corresponding to ppGpp? Can two different compounds peak at the same retention time, and if so, how to distinguish ppGpp from that GTP 92% peak? Also, can you speculate about the nature of the middle (6.2/6.3s retention time) peak, that you refer to as the "contaminant" (line 68)? Given that it is present in most in vivo samples and especially in eggs (Fig. S2), it may have a biological significance.
2. Lines 70, 101, 171, 194. Please add a brief conclusion to these sections
3. Line 71. Please add a rationale as to why you performed these experiments
4. Lines 79 - 80. Considering that you are the first to observe ppGpp in *D.melanogaster*, you do not know whether ppGpp tends to be degraded or "induced" at any given stage by Mesh1. This remains to be demonstrated, but you can present this as a hypothesis. Note that low levels of Mesh1 would not induce ppGpp, but rather would prevent ppGpp degradation or lead to ppGpp accumulation.
5. Fig. 2D. The error bars overlap, meaning there is very little statistical difference (if any) between the two samples, which is confirmed by the p-value of 0.21. The authors should temper their conclusion to this result.
6. Line 119. While GTP levels and ppGpp levels both change (suggesting a correlation), there is no direct proof here that one influences the other. You can only present this as a hypothesis, related to what is observed in Gram + bacteria where high ppGpp leads to depletion in GTP pools.
7. Line 128-130; Lines 139-141, Line 194, Lines 197-198. There is an effect of Mesh1, however you have not demonstrated that this effect is related to the ppGpp hydrolase activity of Mesh1. This effect could result from the NADPH phosphatase activity of Mesh1, or from any other interaction between Mesh1 and other cellular components.
8. Fig. S4B; also lines 170-171. One would expect ppGpp to accumulate in Mesh1 lof specifically under starvation conditions, ie when ppGpp is expected to be produced while the hydrolase is absent. Could you elaborate on that, perhaps in the discussion? Maybe various hydrolases exist that function under different conditions, in different organs and/or at different stages of development (also suggested by the absence of phenotype of Mesh1 lof on the eyes)?
9. Fig. 6. Could you highlight the pathways that are expected to be affected by NADPH levels, and hence by the phosphatase activity of Mesh1? This could help extrapolating what effects could be more specifically related to the ppGpp hydrolase activity.
10. Lines 203-204. This is not accurate; you have no evidence of any direct regulation by ppGpp. Again, these changes only demonstrate an effect of Mesh1, that may or may not require the ppGpp second messenger. (Though beyond the scope of this study and not expected from the authors in this review, the use of mutants of the hydrolase domain and of the phosphatase domain of Mesh1 would help sort this out, for instance.)

Minor comments

- Line 63. Typo "ssubjected"
- Fig. S3B. Define "a" and "b" in the legends
- Line 106. Define "lof" like you did with "gof"

- Fig. S2E/F. Results for Mesh1 lof strains should ideally be presented next to wt strains, for comparison purposes.
- Line 136. Could it be more accurate to say that all ppGpp levels were normal except for Mesh1 lof in high nutrient, where ppGpp spiked?
- Line 163. Typo "beforeinduction"

Reviewer #3 (Remarks to the Author):

The manuscript submitted by Ito et al. focuses on the detection of the stringent response alarmone – ppGpp – in metazoa (namely *Drosophila* and a human cell line), and the consequences of ppGpp accumulation in terms of metabolism, cell death and organism survival. It represents the first report of ppGpp detection in animals (there have been numerous previous attempts). It also builds on existing research regarding the stringent response in *Drosophila* through genetic manipulation of the known ppGpp hydrolase, Mesh1.

Overall, the manuscript is well written and well organized, and addresses a decades-old enigma: the presence/absence of ppGpp in animals. However, given that the detection of ppGpp in both *Drosophila* and a human cell line is presented in both the abstract and discussion as the key, take-home message from this study, I feel that: a) the apparent experimental detection of ppGpp requires corroboration; and b) insufficient background and discussion has been provided on the numerous previous attempts to detect ppGpp in animal cells, and how/why the current study has apparently been successful where others have failed.

Specific comments:

1. The authors used MS-MS to detect ppGpp in cellular extracts. I appreciate that the authors have provided a number of lines of evidence which suggest that the peak detected represents ppGpp, including spiking samples with a ppGpp standard, and comparing the amount of the detected molecule in wildtype, *Mesh1* gain-of-function and *Mesh1* loss-of-function *Drosophila*. However, given that this is the first report of ppGpp detection in animals and no putative metazoan ppGpp synthetase has been identified (not even bioinformatically - ref. 14), the identity of the peak needs to be corroborated through at least one other method. ppGpp has previously been detected and quantified in bacterial cell extracts using ³¹P-NMR spectroscopy (de la Fuente-Nunez et al. 2014, PLoS Pathog. e1004152) or a ppGpp-specific fluorescent chemosensor (Rhee et al. 2008, J Am Chem Soc. 130:784).
2. The authors cite only a single (recent) study in which an attempt was made to detect ppGpp in animal cells (ref. 21). However, the search for ppGpp in metazoa dates back to the 1970s, when a number of attempts were made (reviewed by Silverman and Atherly, 1979, Microbiol Rev, 43:27). Presumably advances in instrumentation have made the present detection of ppGpp possible, but no discussion of the methods used by others is given and, indeed, the final sentence of the discussion ("Overall, our new method for quantifying ppGpp in animal cells should provide new opportunities for investigating the stringent response in animals") is not currently supported by any description/discussion of this "new" method.
3. Sun et al. (ref. 21) have previously reported gene expression data for a *Drosophila Mesh1* null mutant in response to amino acid starvation. How do the authors' metabolomic data compare with these gene expression data?
4. Further, Sun et al. also reported on the phenotype of a *Drosophila Mesh1* null mutant. The present findings on the phenotype of *Mesh1* loss-of-function *Drosophila* should be discussed in the context of the Sun et al. study.

Responses to the reviewers' comments

Thank you very much for your thoughtful comments and recommendations. We have revised the manuscript based on your comments and recommendations. Descriptions of specific passages we revised are provided below, and the revised text is shown in red in the manuscript.

Reviewer #1:

In this paper, the authors detect ppGpp in *Drosophila melanogaster* during different developmental stages. The use of germ-free *Drosophila* larvae and antibiotic treated cultured cells suggest that the ppGpp detected is not all from bacterial origin. Similar results were obtained with human cells. Varying the levels of Mesh-1 hydrolase by overexpression or deletion, produce the expected changes on the ppGpp levels, that are accompanied by metabolic changes. The authors show that high levels of ppGpp produce cellular lethality. Several attempts have been made to detect ppGpp in metazoan cells but without much success until now. Ito et al. seem to have succeeded on detecting ppGpp by LC-ESI-MS/MS in metazoa.

1. RSH enzymes usually are classified as long RSH enzymes, that contain domains for synthesis and hydrolysis (such as SpoT), and small alarmone synthetase (SAS, like Yjbm) or small alarmone hydrolases (SAH, like Mesh-1). For clarity, this could be mentioned in the introduction or figure 1A. Also, to unify nomenclature between different species, Yjbm is referred as RelQ by several authors.

OUR ANSWER: We agree with the reviewer's comment. We added sentences mentioning the RSH classification in the Introduction in the revised manuscript (line 41~51). We also added a sentence indicating that YjbM corresponds to RelQ (line 47~48).

2. In figure 1B and C, the authors detect a peak for GTP, ppGpp and then an unknown peak in between, present also in the ppGpp standard. Can it be pGpp? Some synthetases, like RelQ from *Enterococcus faecalis* produces pGpp (doi 10.1128/JB.00324-15). Also ppGpp and pppGpp can be degraded to pGpp in *B. subtilis* by a NuDiX enzyme (<https://doi.org/10.1101/2020.03.23.003749>). pGpp has a differential motility than GTP in TLC and LC-MS, being usually found between GTP and ppGpp, as the unknown compound. It could be determined by running a pGpp standard as done by ppGpp alone. Considering that the levels of the unknown peak changes between the differential stages of *Drosophila* (figure S2) and between different human cell cultures (figure 1C), seem relevant to determine if this unknown peak is pGpp or not.

OUR ANSWER: Regarding the unknown peak, it must be a molecule having a molecular mass of 602 (the same as that of ppGpp), which could pass through the first selected MS (MS1), and

also have a diphosphate moiety with a molecular mass of 159, that could be detected in the second fragmented MS (MS2) in the tandem mass spectrometer used for the analysis. This means that the unknown peak must not be pGpp or pppGpp. Please note that we simultaneously monitored ppGpp and GTP in MS1 using the Multiple-Reaction-Monitoring mode, so small amounts of GTP unavoidably contaminated the ppGpp fraction ($m/z = 602$). Because the amount of GTP is much higher than that of ppGpp, and diphosphate was released by fragmentation of GTP as with ppGpp, the GTP peak appeared in the $m/z = 602 \rightarrow 159$ detection mode (guanosine tetraphosphate detection mode). This phenomenon is called Cross-Talk in the MS/MS analysis.

To get more information about the unknown peak molecule, we additionally performed MS/MS fragmentation analysis of the unknown peak and found that its structure must be very similar to that of ppGpp because the MS/MS fragmented pattern of the unknown peak was almost identical to that of ppGpp (Supplementary Fig. S1). We assumed that the unknown peak is likely to be guanosine 5'-diphosphate, 2'-diphosphate (ppGpp is guanosine 5'-diphosphate, 3'-diphosphate). To determine the exact structure of the unknown peak, other analyses such as NMR must be done; however, because the amount of the unknown peak is very low even in the GTP and ppGpp standards, it is essentially impossible to determine the structure. Please note that we have to monitor fragmented ions after LC, so we could not collect the non-fragmented sample after MS/MS detection, as is done with HPLC-based detection. We hope that the reviewer can understand our situation. We added sentences mentioning the new data in the revised manuscript (line 94~100, 105~110, 283~287; Supplemental Fig. S1).

We agree that Nudix enzymes can degrade ppGpp. This possibility is newly discussed in the revised manuscript (line 303~307). We thank the reviewer for pointing this out.

3. Together with ppGpp, many synthetases produce also pppGpp, although it is not mentioned in the paper. Can pppGpp be detected with this method? Or is pppGpp not detected in *Drosophila*? This should be discussed in the paper as either deficiency of the method or as a differential characteristic.

OUR ANSWER: We agree with the comment that readers may want to see how we can detect pppGpp by our method. In fact, we have tried to detect pppGpp; however, we have not yet succeeded in clearly detecting a pppGpp elution peak in the LC-MS/MS, perhaps due to low affinity of pppGpp for the column in the solvent used for ppGpp separation. This information is now included in the revised manuscript (lines 116~124). Methods for pppGpp standard preparation are also provided in the methods section (line 538~549).

4. The figure 2 seem to show divergences on the levels of ppGpp in pupae stage between panels

A and D. In figure 2A the levels of ppGpp in pupae are up to 250 pmol/g FW while in figure 2D seems to be up to 600 pmol/g FW. Are the levels of ppGpp of *Mesh1* *gof* from figure 2D being compared to w- (as indicated in the legend) or to yw; *Act5c-Gal4* (as suggested in the methods section)? If they are compared to yw, please, indicate it in the figure legend. Then, why are the levels of ppGpp in yw pupae higher than w-? Please, add the ppGpp measurements of yw pupae in the table S1 too.

OUR ANSWER: We agree with the comment that it is unclear why the amount of ppGpp in the control of *Mesh1 GOF* were higher than in w-. The reason was unclear, but may be due to different genotypes. This point is now described in the text (line 178~181). In fact, we used w/yw; +/+; +/- *Act5c-Gal4* as the control, and this information is included in the revised manuscript (Fig. 2D).

5. The results shown in figure 3 seem a bit challenging and raised the following concerns:

a. The authors show hatching rates from larvae to pupae and from pupae to adult in figure 3A, but they talk about the final hatching rate from larvae to adult in the text. At the same time, they only show ppGpp levels on larvae (fig. 3B). From my point of view (with not much knowledge in *Drosophila*), wouldn't be better just focusing on the hatching rate from larvae to pupae? If not, were the ppGpp levels measured during pupae stage during starvation conditions? It seems to me that to discuss properly the effects of ppGpp on the hatching from larvae to adult, the levels of ppGpp should be determined in pupae stage too, not only in larvae as shown in fig. 3B.

OUR ANSWER: We agree with the reviewer's comment. In the revised manuscript, we deleted sentences mentioning results of hatching rate from pupae to adults, and we only focused on the results of hatching rate from larvae to pupae (line 189~192). Furthermore, we describe only the hatching rate in *Mesh1* LOF from larvae to pupae (line 192~195).

b. The authors show that there is a decrease on the hatching rate in wt flies between control and starvation conditions (72% to 49%), but no differences in the ppGpp levels are detected. To me, these data suggest that ppGpp is not increased during the starvation conditions used in this study and that it doesn't seem responsible for the hatching rate decrease observed in wt flies.

OUR ANSWER: We agree with the reviewer's comment. In the revised manuscript, we clearly refer the fact that ppGpp is not increased upon starvation (line 205~206). However, it is still possible that ppGpp is accumulated in small specific areas of tissues upon starvation (line 298~299).

c. The authors performed the same experiment with *Mesh1 lof*. In control conditions, the

authors show an increase on the levels of ppGpp (7-fold) in *Mesh1 lof* and an increase on the hatching rate (72% to 87%). However, when they determine the levels of ppGpp in *Mesh1 lof* starved larvae, they see low levels of ppGpp, similar to the ones observed in wt larvae. No explanation is given for this decrease. Also, they observe a decrease on the hatching rate compared to wt (48% to 37%) during starvation although having similar ppGpp levels. The authors suggest that the pupation delay of *Mesh1 lof* (during starvation) is due a decrease on the GTP levels (figure S3D). The results in figure S3D are performed in control conditions, where the lower levels of GTP are due the high levels of ppGpp, as suggested by the authors. Why the GTP levels of *Mesh1 lof* during control conditions are used to explain the challenging results observed during starvation conditions?

OUR ANSWER: We agree with the reviewer's comment. We changed the original sentence "These results suggested that the lower GTP level in *Mesh1 LOF* was a consequence of an increase in ppGpp, and the lower GTP level under normal conditions induced the pupation delay" to "These results suggested that the pupation delay of *Mesh1 LOF* is irrespective to ppGpp level itself, but potentially due to the lowered GTP level in the mutant" in the revised manuscript (line 208~210).

To me, the results in figure 3 seem to contradict the authors when they suggest that "control of ppGpp concentration is important for growth and the response to starvation". At the same time, the *mesh1 lof* results during starvation remain unexplained. I encourage the authors to clarify these results from figure 3. Alternatively, I would recommend eliminating the *mesh1 lof* results and only discuss the results observed in WT.

OUR ANSWER: We agree with the reviewer's comment. In the revised manuscript, we deleted the sentence and replaced it with "This result is consistent with a previous study that reported that *Mesh1 LOF* larvae showed retarded growth" (line 192~194).

6. In figure 1C, the ppGpp levels of different culture cells and germ-free (GF) flies are determined. The authors show a chromatogram for GF and another for GF control. What is the difference between GF and GF control? Please, indicate it in the figure legend.

OUR ANSWER: We added information about the "GF control", which is adult flies grown under normal "non-GF" conditions (line 791~792). We deleted the chromatogram of larvae, since it is not necessary.

7. In figure 2A and S3A, blue bars represent males or mixed population. I understand that sex cannot be assessed during eggs or pupae stage, resulting in a mixed population. I assume that in

the other stages blue bars refer to male population. Please, use different colors to differentiate between mixed populations and males.

OUR ANSWER: We agree, and changed the colors of the figures to distinguish male and mixed samples in the revised manuscript (Fig. 2A and S3A).

8. In figures 3, S3 and S4 the statistical significance of the differences is indicated by using letters, but no other explanation is given. What does the different letters mean?

OUR ANSWER: We provided information to define the different letters in each figure legend in the revised manuscript (legends of Fig. 3, Fig. S3, 4).

9. In Hesketh et al. 2017 (DOI: 10.1128/mBio.01047-17) the authors overexpressed RelA in *S. cerevisiae* to determine the effects of ppGpp over gene expression, and they determined that ppGpp upregulates transcription of genes encoding key mitochondrial proteins. It would be interesting to compare their results with the metabolic changes observed in *Drosophila*.

OUR ANSWER: We agree with the reviewer's comment, and added text to discuss the data of this and of the mBio paper in the Discussion section (line 333~355). It is suggested that ppGpp (and Mesh1) may control some specific metabolism including the pentose phosphate pathway. We thank the reviewer for this suggestion.

10. As mentioned by the authors in the discussion, Mesh1 seems to have a NADPH phosphatase activity. Do the authors see such effect in Mesh1 lof or Mesh1 gof? In table S2, there seems to be a 2-fold decrease on the levels of NADP in Mesh1 lof compared to WT. It would be interesting to discuss this on the paper.

OUR ANSWER: We agree with the reviewer's comment, and have added text to point out the fact that NADP levels in Mesh1 LOF are 2-fold lower than that in WT, which is actually not expected, since if Mesh1 has NADPH phosphatase activity, the NADP(H) level is expected to be increased in Mesh1 LOF (line 324~332). We assumed that Mesh1 LOF reduced the NADP(H) pool size to compensate for the loss of NADPH phosphatase activity of Mesh1.

Reviewer #2:

Ito et al propose that guanosine tetraphosphate (ppGpp) is present in measurable levels in *Drosophila melanogaster*, and that this ppGpp molecule would be hydrolyzed in vivo by the recently discovered RelA-SpoT homologue (RSH) Mesh1 hydrolase in certain conditions. They further propose that changes in ppGpp levels could be responsible for multiple Mesh1-mutant derived phenotypes and changes in the metabolome, including starvation phenotypes,

reminiscent of ppGpp-dependent starvation phenotypes observed in bacteria and plants. The main discovery of this study is the first direct observation of the ppGpp molecule in *Drosophila melanogaster* and more generally in metazoa. The authors observed ppGpp at various stages of development, under various growth conditions and also tested germ-free (GF) strains to insure that ppGpp was not produced by commensal bacteria. The authors also demonstrate that Mesh1 is necessary, at least in certain conditions, to maintain low ppGpp levels, hence providing insights into the role of Mesh1 as a ppGpp hydrolase in vivo. As for the diverse phenotypes observed in Mesh1 mutants, while they somewhat correlate with ppGpp levels in certain cases, it is not clear nor demonstrated whether the effect observed results from Mesh1 ppGpp hydrolase activity or its known NADPH phosphatase activity, or yet other functions yet to be discovered of this enzyme. Expression of a heterologous ppGpp synthase provokes a strong phenotype, along with a spike in ppGpp levels, supporting the idea that high ppGpp levels are cytotoxic, but these effects are not necessarily directly related to the effects observed with Mesh1 mutants. Altogether, this manuscript is a valuable contribution to the literature. However, several passages require more detailed explanations and proper conclusions, while several conclusions about the effects of ppGpp in vivo require to be toned down, as for the most part no demonstration of the direct effect of ppGpp is provided.

Major comments

1. Line 68 and Fig. S1. How the authors explain a peak in GTP 92% corresponding to ppGpp? Can two different compounds peak at the same retention time, and if so, how to distinguish ppGpp from that GTP 92% peak? Also, can you speculate about the nature of the middle (6.2/6.3s retention time) peak, that you refer to as the “contaminant” (line 68)? Given that it is present in most in vivo samples and especially in eggs (Fig. S2), it may have a biological significance.

OUR ANSWER: In our LC-MS/MS-based detection method, LC-eluted molecules having molecular mass of 602 (MW ppGpp) and 522 (MW of GTP) were selected by the first MS (MS1) followed by fragmentation, and then fragmented molecule(s) having a molecular mass of 159 (MW of diphosphate moiety) was selectively detected by the second MS (MS2) to monitor ppGpp ($m/z = 602 \rightarrow 159$) and GTP ($m/z = 522 \rightarrow 159$) (as indicated in the inset of Fig. S1). Thus, the unknown and ppGpp peaks could be detected only in the ppGpp detection mode ($m/z = 602 \rightarrow 159$), but not that of GTP ($m/z = 522 \rightarrow 159$). Please note that because GTP is significantly abundant compared to ppGpp, and we monitored ppGpp and GTP at the same time using the Multiple-Reaction-Monitoring mode, a small amount of GTP unavoidably contaminated the ppGpp fraction. Given that both GTP and ppGpp have a diphosphate moiety,

contaminating GTP could be detected by MS2 even in the ppGpp detection mode ($m/z = 602 \rightarrow 159$). This phenomenon is called “cross-talk” of Multiple-Reaction-Monitoring mode in the MS/MS system, as mentioned in the legend of Fig. S1. We assigned the third peak of GTP as the ppGpp elution based on the fact that 1) it is selectively detected by the guanosine tetraphosphate detection mode, 2) it matches the elution time of the ppGpp standard, and 3) its level is lower in the >99% GTP standard than in the 92% GTP standard. From same assumptions, we judged that the unidentified molecule is likely a guanosine tetraphosphate with unknown structure having a molecular mass of 602 (same as that of ppGpp), which could pass through the first selected MS (MS1), and also has a diphosphate moiety, having a molecular mass of 159, that could be detected in the second fragmented MS (MS2) in the tandem mass spectrometer. For revision of the manuscript, we additionally performed MS/MS fragmentation analysis of the unknown peak, and found that the structure of the unknown molecule must be very similar to that of ppGpp because the MS/MS fragment pattern of the unknown peak was almost the same as that of ppGpp (Supplementary Fig. S1). We added text mentioning the new data in the revised manuscript (line 94~100; 105~110; Supplemental Fig. S1). We assumed that the unknown peak is likely to be guanosine 5'-diphosphate, 2'-diphosphate (ppGpp is guanosine 5'-diphosphate, 3'-diphosphate). To determine the exact structure of the unknown peak, other analyses such as NMR must be done; however, because of the very low amount of the unknown peak even in the GTP standard, it is essentially impossible to determine the structure. Please note that we detected the fragmented molecule after LC, so we could not collect the sample after MS/MS detection, as is done with HPLC. We added new text to discuss this possibility in the revised manuscript (line 283~287).

2. Lines 70, 101, 171, 194. Please add a brief conclusion to these sections

OUR ANSWER: We agree, and provided brief conclusions to these sections (line 113~115, 160~161, 239~241, 266~267).

3. Line 71. Please add a rationale as to why you performed these experiments

OUR ANSWER: We agree with the reviewer's comment, and provided a sentence mentioning the purpose of the experiment in the revised manuscript (line 125~126).

4. Lines 79 - 80. Considering that you are the first to observe ppGpp in *D.melanogaster*, you do not know whether ppGpp tends to be degraded or “induced” at any given stage by Mesh1. This remains to be demonstrated, but you can present this as a hypothesis. Note that low levels of Mesh1 would not induce ppGpp, but rather would prevent ppGpp degradation or lead to ppGpp accumulation.

OUR ANSWER: We agree with the reviewer's comment that it is still unknown whether ppGpp tends to be degraded or "induced" at any given stage by Mesh1. We added a sentence to note that "~ it is unknown whether ppGpp in *Drosophila* tends to be degraded or induced at each stage for keeping its appropriate level (line 138~139)."

5. Fig. 2D. The error bars overlap, meaning there is very little statistical difference (if any) between the two samples, which is confirmed by the p-value of 0.21. The authors should temper their conclusion to this result.

OUR ANSWER: We agree with the reviewer's comment. In the revised manuscript we tempered our interpretation to state that "~suggesting that ppGpp degradation was enhanced in *Mesh1 GOF* compared to that in the control strain, although no clear significance was observed ($P = 0.21$, *t*-test)." (line 176~178).

6. Line 119. While GTP levels and ppGpp levels both change (suggesting a correlation), there is no direct proof here that one influences the other. You can only present this as a hypothesis, related to what is observed in Gram + bacteria where high ppGpp leads to depletion in GTP pools.

OUR ANSWER: We agree with the reviewer's comment that this was just hypothesis, and our previous description seemed not to be appropriate. In the revised manuscript, we changed the description to read: "Given GTP synthesis is suppressed by ppGpp in Gram-positive bacteria, ppGpp may play a role in GTP homeostasis in metazoa, although GTP levels in *Mesh1 GOF* pupae were the same as those in WT~" (line 184~186).

7. Line 128-130; Lines 139-141, Line 194, Lines 197-198. There is an effect of Mesh1, however you have not demonstrated that this effect is related to the ppGpp hydrolase activity of Mesh1. This effect could result from the NADPH phosphatase activity of Mesh1, or from any other interaction between Mesh1 and other cellular components.

OUR ANSWER: We agree, and deleted the sentences in the revised manuscript.

8. Fig. S4B; also lines 170-171. One would expect ppGpp to accumulate in *Mesh1 lof* specifically under starvation conditions, ie when ppGpp is expected to be produced while the hydrolase is absent. Could you elaborate on that, perhaps in the discussion? Maybe various hydrolases exist that function under different conditions, in different organs and/or at different stages of development (also suggested by the absence of phenotype of *Mesh1 lof* on the eyes)?

OUR ANSWER: We agree with the reviewer's comment, and added the following text to the Discussion "~ ppGpp synthesis is upregulated, if any, only in small area of specific organs

under starvation conditions” (line 298~299). Furthermore, the existence of other ppGpp hydrolases (such as Nudix) is also mentioned in the revised manuscript (line 299~307).

9. Fig. 6. Could you highlight the pathways that are expected to be affected by NADPH levels, and hence by the phosphatase activity of Mesh1? This could help extrapolating what effects could be more specifically related to the ppGpp hydrolase activity.

OUR ANSWER: Together with a comment given by the reviewer 1 suggesting that we compare results of this and ppGpp-accumulating yeast (Hesketh et al. 2017 (DOI: 10.1128/mBio.01047-17), we further analyzed our metabolomics data, which additionally suggested that Mesh1 may control some specific metabolism including the pentose phosphate pathway by controlling ppGpp levels and NADP(H) pool size. A relevant discussion is now included in the revised manuscript (line 340~355). Furthermore, we newly added the GSH-GSSG pathway in which NADPH and NADP levels were affected in the Mesh1 *LOF* (Fig. 6).

10. Lines 203-204. This is not accurate; you have no evidence of any direct regulation by ppGpp. Again, these changes only demonstrate an effect of Mesh1, that may or may not require the ppGpp second messenger. (Though beyond the scope of this study and not expected from the authors in this review, the use of mutants of the hydrolase domain and of the phosphatase domain of Mesh1 would help sort this out, for instance.)

OUR ANSWER: We agree with the reviewer’s comment. In the revised manuscript, we changed the sentence mentioning that “~ physiological role of the Mesh1-dependent ppGpp control in matazoa is still largely unknown, our metabolome analyses suggest that enzymes involved in purine metabolism are regulated by the Mesh1 activity (line 308~310).

Minor comments

• Line 63. Typo “ssubjected”

OUR ANSWER: We corrected the wording in the revised manuscript (line 93).

• Fig. S3B. Define “a” and “b” in the legends

OUR ANSWER: we added the information to the legend (legend of Fig. S3).

• Line 106. Define “lof” like you did with “gof”

OUR ANSWER: “lof” is already defined before the indicated sentence (line 53), and we also used “lof” in a later sentence (line 167).

- Fig. S2E/F. Results for Mesh1 lof strains should ideally be presented next to wt strains, for comparison purposes.

OUR ANSWER: We agree with the reviewer's comment. We added WT chromatograms in Fig. S2E/F in the revised manuscript.

- Line 136. Could it be more accurate to say that all ppGpp levels were normal except for Mesh1 lof in high nutrient, where ppGpp spiked?

OUR ANSWER: In the revised manuscript, we changed the sentence to say that “~ ppGpp levels in *w* were normal under starvation conditions, but it specifically increases in rich medium in *Mesh1 LOF*” (line 205~206).

- Line 163. Typo “beforeinduction”

OUR ANSWER: We corrected the wording in the revised manuscript (line 231)

Reviewer #3:

The manuscript submitted by Ito et al. focuses on the detection of the stringent response alarmone – ppGpp – in metazoa (namely *Drosophila* and a human cell line), and the consequences of ppGpp accumulation in terms of metabolism, cell death and organism survival. It represents the first report of ppGpp detection in animals (there have been numerous previous attempts). It also builds on existing research regarding the stringent response in *Drosophila* through genetic manipulation of the known ppGpp hydrolase, Mesh1.

Overall, the manuscript is well written and well organized, and addresses a decades-old enigma: the presence/absence of ppGpp in animals. However, given that the detection of ppGpp in both *Drosophila* and a human cell line is presented in both the abstract and discussion as the key, take-home message from this study, I feel that: a) the apparent experimental detection of ppGpp requires corroboration; and b) insufficient background and discussion has been provided on the numerous previous attempts to detect ppGpp in animal cells, and how/why the current study has apparently been successful where others have failed.

Specific comments:

1. The authors used MS-MS to detect ppGpp in cellular extracts. I appreciate that the authors have provided a number of lines of evidence which suggest that the peak detected represents ppGpp, including spiking samples with a ppGpp standard, and comparing the amount of the

detected molecule in wildtype, *Mesh1* gain-of-function and *Mesh1* loss-of-function *Drosophila*. However, given that this is the first report of ppGpp detection in animals and no putative metazoan ppGpp synthetase has been identified (not even bioinformatically - ref. 14), the identity of the peak needs to be corroborated through at least one other method. ppGpp has previously been detected and quantified in bacterial cell extracts using ^{31}P -NMR spectroscopy (de la Fuente-Nunez et al. 2014, PLoS Pathog. e1004152) or a ppGpp-specific fluorescent chemosensor (Rhee et al. 2008, J Am Chem Soc. 130:784).

OUR ANSWER: As for the request to verify the identity of the ppGpp peak by at least one other method, we actually have tried to do that. However, we found that because the amount of ppGpp present in animal cells is very low, NMR analysis is not realistic. As for the fluorescent chemosensor, data in the JACS paper indicate that the sensitivity of the chemosensor is in the micromolar range, so that it is not sensitive enough to detect ppGpp isolated from animal tissues. Please also note that we detected a fragmented molecule derived from ppGpp, so we could not collect separate samples after LC-MS/MS for further analysis. In conclusion, it is technically still impossible to perform determine ppGpp peak analysis with other currently available methods. However, to further confirm whether the detected peak corresponds to ppGpp, we performed additional experiments to show multiple fragmented ions derived from the elution and the ppGpp standard. The features of the fragmented ions were almost identical between ppGpp standard, and the possible ppGpp elution peak from the *Drosophila* extract (Supplementary Fig. S1), strongly suggesting that the elution peak represents ppGpp. This data is included in the revised manuscript (Supplementary Fig. S1; line 94~100).

2. The authors cite only a single (recent) study in which an attempt was made to detect ppGpp in animal cells (ref. 21). However, the search for ppGpp in metazoa dates back to the 1970s, when a number of attempts were made (reviewed by Silverman and Atherly, 1979, Microbiol Rev, 43:27). Presumably advances in instrumentation have made the present detection of ppGpp possible, but no discussion of the methods used by others is given and, indeed, the final sentence of the discussion (“Overall, our new method for quantifying ppGpp in animal cells should provide new opportunities for investigating the stringent response in animals”) is not currently supported by any description/discussion of this “new” method.

OUR ANSWER: We agree with the reviewer’s comment. In the revised manuscript, we provide more details of previous studies in which different researchers attempted to detect ppGpp from animal cells (line 58~69). We also provide more information about how we modified our previous method to detect ppGpp in animal tissues in the revised manuscript (line 274~280).

3. Sun et al. (ref. 21) have previously reported gene expression data for a *Drosophila*

Mesh1 null mutant in response to amino acid starvation. How do the authors' metabolomic data compare with these gene expression data?

OUR ANSWER: We agree with the reviewer's comment that gene expression data from Sun et al. is useful to discuss our and previous studies. In the revised manuscript, we refer to the Sun et al. data in the discussion section (line 340~343). We thank the reviewer for this suggestion.

4. Further, Sun et al. also reported on the phenotype of a *Drosophila Mesh1* null mutant. The present findings on the phenotype of *Mesh1* loss-of-function *Drosophila* should be discussed in the context of the Sun et al. study.

OUR ANSWER: We agree with the reviewer's comment. In the revised manuscript, we added several sentences discussing our and other previous (Sun et al) data in the revised manuscript (line 192~194; 359~367).

REVIEWERS' COMMENTS:

Reviewer #2 (Remarks to the Author):

Ito et al propose that guanosine tetraphosphate (ppGpp) is present in measurable levels in *Drosophila melanogaster*, and that this ppGpp molecule would be hydrolyzed in vivo by the recently discovered RelA-SpoT homologue (RSH) Mesh1 hydrolase in certain conditions. They further propose that changes in ppGpp levels could be responsible for multiple Mesh1-mutant derived phenotypes and changes in the metabolome, including starvation phenotypes, reminiscent of ppGpp-dependent starvation phenotypes observed in bacteria and plants.

The main discovery of this study is the first direct observation of the ppGpp molecule in *Drosophila melanogaster* and more generally in metazoa. The authors observed ppGpp at various stages of development, under various growth conditions and also tested germ-free (GF) strains to insure that ppGpp was not produced by commensal bacteria. Several Mesh1 mutants are studied and provide additional information on the effect of this enzyme, and hence possibly of ppGpp, in *Drosophila*. Various phenotypes and effects are observed on both development, cell death and the metabolome of *Drosophila melanogaster*.

Overall, this study is of great interest, and the authors have provided useful explanations and made commendable efforts to improve the manuscript when needed. No further modifications are required.

Reviewer #3 (Remarks to the Author):

I appreciate the difficulties the authors have faced with regard to corroborating the identity of the ppGpp peak via other methods due to its low abundance in animal cells, and do not feel that this challenge should prevent publication of their findings. However, the authors must explicitly acknowledge the limitations of their data and that corroboration with an independent method is not presently possible; I suggest this acknowledgment is inserted around line 100 and the difficulties with other methods added to the discussion. In line with this, the wording on lines 96 and 99 needs to be toned down; given the lack of independent corroboration, "indicating" and "confirming" are not appropriate language to be used here and should be replaced (e.g. "strongly suggest").

Responses to the reviewers' comments

Thank you very much for reviewing our revised manuscript. We have further modified the manuscript based on your comments and recommendations. Descriptions of specific passages we revised are provided below.

Reviewer #2:

Ito et al propose that guanosine tetraphosphate (ppGpp) is present in measurable levels in *Drosophila melanogaster*, and that this ppGpp molecule would be hydrolyzed *in vivo* by the recently discovered RelA-SpoT homologue (RSH) Mesh1 hydrolase in certain conditions. They further propose that changes in ppGpp levels could be responsible for multiple Mesh1-mutant derived phenotypes and changes in the metabolome, including starvation phenotypes, reminiscent of ppGpp-dependent starvation phenotypes observed in bacteria and plants.

The main discovery of this study is the first direct observation of the ppGpp molecule in *Drosophila melanogaster* and more generally in metazoa. The authors observed ppGpp at various stages of development, under various growth conditions and also tested germ-free (GF) strains to insure that ppGpp was not produced by commensal bacteria. Several Mesh1 mutants are studied and provide additional information on the effect of this enzyme, and hence possibly of ppGpp, in *Drosophila*. Various phenotypes and effects are observed on both development, cell death and the metabolome of *Drosophila melanogaster*.

Overall, this study is of great interest, and the authors have provided useful explanations and made commendable efforts to improve the manuscript when needed. No further modifications are required.

OUR ANSWER: The reviewer did not give further request for publication of the manuscript.

Reviewer #3:

I appreciate the difficulties the authors have faced with regard to corroborating the identity of the ppGpp peak via other methods due to its low abundance in animal cells, and do not feel that this challenge should prevent publication of their findings. However, the authors must explicitly acknowledge the limitations of their data and that corroboration with an independent method is not presently possible; I suggest this acknowledgment is inserted around line 100 and the difficulties with other methods added to the discussion. In line with this, the wording on lines 96 and 99 needs to be toned down; given the lack of independent corroboration, “indicating” and “confirming” are not appropriate language to be used here and should be replaced (e.g. “strongly suggest”).

OUR ANSWER: We agree for the comment, so that we inserted a sentence mentioning the difficulties to further confirm ppGpp elution by other methods, in the revised manuscript (Lines 96~100). We also changed the wording “indicating” to “strongly suggesting” in the revised manuscript, as suggested (Line 95~96 and 103).